 

# Physical basis of large microtubule aster growth

Keisuke Ishihara[1,2*†‡], Kirill S Korolev[3*], Timothy J Mitchison[1,2]

[1]Department of Systems Biology, Harvard Medical School, Boston, United States; [2]Cell Division Group, Marine Biological Laboratory, Woods Hole, United Sates; [3]Department of Physics and Graduate Program in Bioinformatics, Boston University, Boston, United States

**Abstract** Microtubule asters - radial arrays of microtubules organized by centrosomes - play a fundamental role in the spatial coordination of animal cells. The standard model of aster growth assumes a fixed number of microtubules originating from the centrosomes. However, aster morphology in this model does not scale with cell size, and we recently found evidence for non-centrosomal microtubule nucleation. Here, we combine autocatalytic nucleation and polymerization dynamics to develop a biophysical model of aster growth. Our model predicts that asters expand as traveling waves and recapitulates all major aspects of aster growth. With increasing nucleation rate, the model predicts an explosive transition from stationary to growing asters with a discontinuous jump of the aster velocity to a nonzero value. Experiments in frog egg extract confirm the main theoretical predictions. Our results suggest that asters observed in large fish and amphibian eggs are a meshwork of short, unstable microtubules maintained by autocatalytic nucleation and provide a paradigm for the assembly of robust and evolvable polymer networks.

*For correspondence: ishihara@ mpi-cbg.de (KI); korolev@bu.edu (KSK)

Present address: †Max Planck Institute of Molecular Cell Biology and Genetics, Dresden, Germany; ‡Max Planck Institute for the Physics of Complex Systems, Dresden, Germany

Competing interests: The authors declare that no competing interests exist.

## Introduction

Animal cells use asters, radial arrays of microtubules, to spatially organize their cytoplasm (*Wilson, 1896*). Specifically, astral microtubules transport organelles (*Grigoriev et al., 2008*; *Wang et al., 2013*; *Waterman-Storer and Salmon, 1998*), support cell motility by mediating mechanical and biochemical signals (*Etienne-Manneville, 2013*), and are required for proper positioning of the nucleus, the mitotic spindle, and the cleavage furrow (*Field et al., 2015*; *Grill and Hyman, 2005*; *Neumüller and Knoblich, 2009*; *Tanimoto et al., 2016*; *Wilson, 1896*). Within asters, individual microtubules undergo dynamic instability (*Mitchison and Kirschner, 1984*): They either grow (polymerize) or shrink (depolymerize) at their plus ends and stochastically transition between these two states. The collective behavior of microtubules is less well understood, and it is not clear how dynamic instability of individual microtubules controls aster growth and function.

The standard model of aster growth posits that centrosomes nucleate and anchor all microtubules at their minus ends while the plus ends polymerize outward via dynamic instability (*Brinkley, 1985*). As a result, aster growth is completely determined by the dynamics of individual microtubules averaged over the growing and shrinking phases. In particular, the aster either expands at a velocity given by the net growth rate of microtubules or remains stationary if microtubules are unstable and tend to depolymerize (*Belmont et al., 1990*; *Dogterom and Leibler, 1993*; *Verde et al., 1992*).

The standard model of aster growth is being increasingly challenged by reports of microtubules with their minus ends located far away from centrosomes (*Akhmanova and Steinmetz, 2015*; *Keating and Borisy, 1999*). Some of these microtubules may arise simply by detachment from centrosomes (*Keating et al., 1997*; *Waterman-Storer et al., 2000*) or severing of pre-existing microtubules (*Roll-Mecak and McNally, 2010*). However, new microtubules could also arise due to a

**eLife digest** Cells must carefully organize their contents in order to work effectively. Protein filaments called microtubules often play important roles in this organization, as well as giving structure to the cell. Many cells contain structures called asters that are formed of microtubules that radiate out from a central point (much like a star shape). Textbooks generally state that all microtubules in the aster grow outward from its center. If this was the case, the microtubules at the edge of large asters – such as those found in frog egg cells and other extremely large cells – would be spread relatively far apart from each other. However, even at the edges of large asters, the microtubules are quite densely packed.

In 2014, a group of researchers proposed that new microtubules could form throughout the aster instead of all originating from the center. This model had not been tested; it was also unclear under what conditions an aster would be able to grow to fill a large cell.

Ishihara et al. – including some of the researchers involved in the 2014 work – have now developed a mathematical theory of aster growth that is based on the assumption that microtubules stimulate the generation of new microtubules. The theory reproduces the key features seen during the growth of asters in large cells, and predicts that the asters may stay at a constant size or grow continuously. The condition required for the aster to grow is simple: each microtubule in it has to trigger the generation of at least one new microtubule during its lifetime. Ishihara et al. have named this process "collective growth".

Experiments performed using microtubules taken from crushed frog eggs and assembled under a cover slip provided further evidence that asters grow via a collective growth process. Future studies could now investigate whether collective growth also underlies the formation of other cellular structures.

nucleation process independent of centrosomes (*Clausen and Ribbeck, 2007*; *Efimov et al., 2007*; *Petry et al., 2013*) and contribute to both aster growth and its mechanical properties. Indeed, we recently observed that centrosomal nucleation is insufficient to explain the large number of growing plus ends found in asters (*Ishihara et al., 2014*). Moreover, the standard model demands a decrease in microtubule density at aster periphery, which is inconsistent with aster morphology in frog and fish embryos (*Wühr et al., 2008*, *2010*). To resolve these inconsistencies, we proposed an autocatalytic nucleation model, where microtubules or microtubule plus ends stimulate the nucleation of new microtubules at the aster periphery (*Ishihara et al., 2014a*, *2014b*; *Wühr et al., 2009*). This mechanism generates new microtubules necessary to maintain a constant density as the aster expands. We also hypothesized that autocatalytic nucleation could effectively overcome extinction of individual microtubules, and allow rapid growth of large asters made of short, unstable microtubules. However, we did not provide a quantitative model that can be compared to the experiments or even show that the proposed mechanism is feasible.

Here, we develop a quantitative biophysical model of aster growth with autocatalytic nucleation. It predicts that asters can indeed expand even when individual microtubules turn over and disappear by depolymerization. In this regime, aster expansion is driven by the increase in the total number of microtubules, and the resulting aster is a network of short interconnected microtubules. The transition from stationary to growing asters depends on the balance between polymerization dynamics and nucleation. At this transition, our theory predicts a minimum rate at which asters grow, which we define as the gap velocity. This gap velocity arises due to the dynamic instability of microtubule polymerization and excludes a wide class of alternative models. More importantly, this mode of aster growth allows the cell to assemble asters with varying polymer densities at consistently large speeds. Using a cell-free reconstitution approach (*Field et al., 2014*; *Nguyen et al., 2014*), we perform biochemical perturbations and observe the slowing down and eventual arrest of aster growth with a substantial gap velocity at the transition. By combining theory and experiments, we provide a quantitative framework for how the cell cycle may regulate the balance between polymerization dynamics and nucleation to control aster growth. We propose that the growth of large interphase asters is an emergent property of short microtubules that constantly turnover and self-amplify.

## Results

### Conceptual model for aster growth based on polymerization dynamics and autocatalytic nucleation

Asters are large structures comprised of thousands of microtubules. How do the microscopic dynamics of individual microtubules determine the collective properties of asters such as their morphology and growth rate? Can asters sustain growth when individual microtubules are unstable? To address these questions, we develop a theoretical framework that integrates polymerization dynamics and

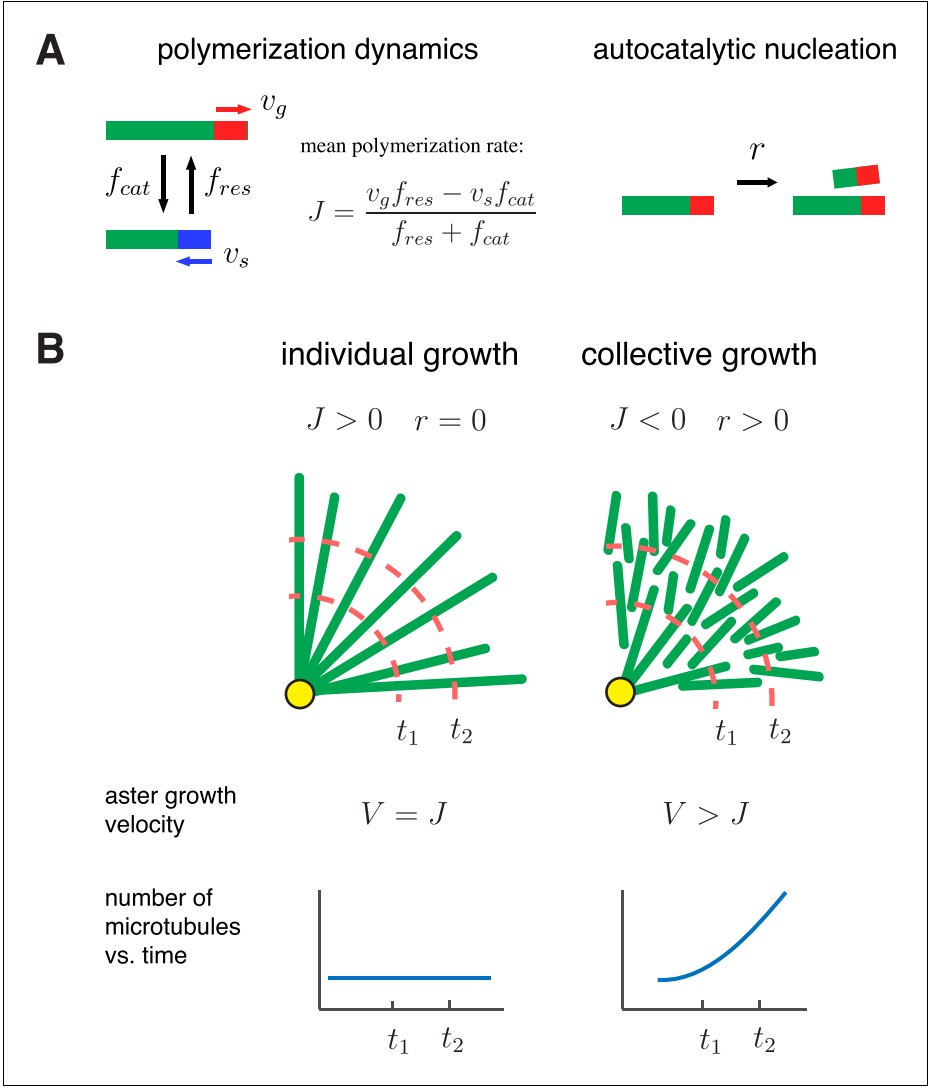

**Figure 1.** A biophysical model for the collective growth of microtubule asters. (**A**) We propose that asters grow via two microscopic processes: polymerization and nucleation. Individual microtubules follow the standard dynamic instability with a growing state with polymerization rate $v_g$ and a shrinking state with depolymerization rate $v_s$. Transitions between the states occur at rates $f_{cat}$ and $f_{res}$, which model catastrophe and rescue events, respectively. New microtubules are added at a rate $r$ via a nucleation at pre-existing plus ends in the growing state. (**B**) Individual vs. collective growth of asters. In the standard model of 'individual growth', asters increase their radius at rate $V = \frac{d\,Radius}{dt}$ only via a net polymerization from the centrosome (yellow). Thus, this model predicts that the rate of aster growth equals the mean polymerization rate $V = J$, the number of microtubules is constant, and their density decreases away from the centrosomes. In the collective growth model, the microtubule density is constant and the number of microtubules increases. Autocatalytic nucleation makes asters grow faster than the net polymerization rate $J$ and can sustain growth even when individual microtubules are unstable $J<0$.

autocatalytic nucleation (**Figure 1A**). Our main goal is to determine the distribution of microtubules within asters and the velocity at which asters grow:

$$V = \frac{d\,Radius}{dt}. \tag{1}$$

Beyond being the main experimental readout, the aster velocity is crucial for cell physiology because it allows large egg cells to divide its cytoplasm rapidly.

Polymerization dynamics of plus ends is an individual property of microtubules. To describe plus end dynamics, we adopt the two-state model of microtubule dynamic instability (**Figure 1A**, left). In this model, a single microtubule is in one of the two states: (i) the growing state, where plus ends polymerize at rate $v_g$ and (ii) the shrinking state, where plus ends depolymerize at rate $v_s$. A growing microtubule may transition to a shrinking state (catastrophe event) with rate $f_{cat}$. Similarly, the shrinking to growing transition (rescue event) occurs at rate $f_{res}$. For large asters growing in *Xenopus* egg cytoplasm, we provide estimates of these parameters in **Table 1**.

Plus end dynamics can be conveniently summarized by the time-weighted average of the polymerization and depolymerization rates (**Dogterom and Leibler, 1993**; **Verde et al., 1992**):

$$J = \frac{v_g f_{res} - v_s f_{cat}}{f_{res} + f_{cat}}. \tag{2}$$

This parameter describes the tendency of microtubules to grow or shrink. When $J<0$, microtubules are said to be in the bounded regime because their length inevitably shrinks to zero, i.e. microtubule disappears. When $J>0$, microtubules are said to be in the unbounded regime, because they have a nonzero probability to become infinitely long. Parameter $J$ also determines the mean elongation rate of a very long microtubule that persists over many cycles of catastrophe and rescue. The dynamics of short microtubules, however, depends on their length and initial state (growing vs. shrinking) and should be analyzed carefully.

The standard model posits that asters are produced by the expansion of individual microtubules, so the transition from small mitotic asters to large interphase asters is driven by a change in the sign of $J$ (**Dogterom and Leibler, 1993**; **Verde et al., 1992**) (**Figure 1B** left, 'individual growth'). With bounded dynamics $J<0$, the standard model predicts that every microtubule shrinks to zero length and disappears. This microtubule loss is balanced by nucleation of new microtubules at the centrosomes, the only place where nucleation is allowed in the standard model. As a result, asters remain in the stationary state and are composed of a few short microtubules, and the aster velocity is thus $V = 0$. With unbounded dynamics $J>0$, the standard model predicts an aster that has a constant number of microtubules and increases its radius at a rate equal to the elongation rate of microtubules (i.e. $V = J$).

**Table 1.** Model parameters used to describe large aster growth reconstituted in interphase *Xenopus* egg extract.

| Quantity | Symbol | Value | Comment |
|---|---|---|---|
| Polymerization rate | $v_g$ | 30 µm/min | Measured from growing plus ends and EB1 comets |
| Depolymerization rate | $v_s$ | 42 µm/min | Measured from shrinking plus ends (**Ishihara et al., 2014a**) |
| Catastrophe rate | $f_{cat}$ | 3.3 min$^{-1}$ | Measured from EB1 comet lifetimes (see Materials and methods) |
| Rescue rate | $f_{res}$ | 2.0±0.3 min$^{-1}$ | Estimated from **Equations (4) and (6)** |
| Autocatalytic nucleation rate | $r$ | 2.1±0.2 min$^{-1}$ | Estimated from **Equations (4) and (6)** |
| Carrying capacity of growing ends | $K$ | 0.4 µm$^{-2}$ | Estimated from comparing $C_g^{bulk}$ to predicted (see SI) |
| Mean microtubule length | $\langle l \rangle$ | 16 ± 2 µm | Estimated from from dynamics parameters (see SI) |
| Aster velocity | $V$ | 22.3±2.6 µm/min | Measured from rate of aster radius increase |
| Gap velocity | $V_{gap}$ | 12.8±1.7 µm/min | Measured from aster growth at 320 nM MCAK-Q710 |
| Bulk growing plus end density | $C_g^{bulk}$ | 0.053±0.030 µm$^{-2}$ | Measured from EB1 comet density (**Ishihara et al., 2014a**) |

Below, we add autocatalytic microtubule nucleation (*Figure 1A*, right) to the standard model and propose the regime of 'collective growth' (*Figure 1B*, right). In this regime, asters grow (*V>0*) although individual microtubules are bounded (*J<0*) and are, therefore, destined to shrink and disappear. The growth occurs because more microtubules are nucleated than lost, and new microtubules are typically nucleated further along the expansion direction. Indeed, when a new microtubule is nucleated, it is in a growing state and starts expanding outward before its inevitable collapse. During its lifetime, this microtubule can nucleate a few more microtubules all of which are located further along the expansion direction. As we show below, this self-amplifying propagation of microtubules is possible only for sufficiently high nucleation rates necessary to overcome microtubule loss and sustain collective growth.

Specifically, we assume that new microtubules nucleate at locations away from centrosomes at rate $Q$. This rate could depend on the local density of growing plus ends if they serve as nucleation sites or the local polymer density if nucleation occurs throughout a microtubule. The new microtubules have zero length and tend to grow radially due to mechanical interactions with the existing microtubule network. These non-centrosomal microtubules disappear when they shrink back to their minus ends. Our assumptions are broadly consistent with known microtubule physiology (*Clausen and Ribbeck, 2007*; *Petry et al., 2013*), and we found strong evidence for nucleation away from centrosomes in egg extract by microtubule counting in growing asters (*Ishihara et al., 2014a*). Below, we consider plus-end-stimulated nucleation and the analysis for the polymer-stimulated nucleation is summarized in the SI.

Without negative feedback, autocatalytic processes lead to exponential growth, but there are several lines of evidence for an apparent 'carrying capacity' of microtubules in a given cytoplasmic volume (*Clausen and Ribbeck, 2007*; *Ishihara et al., 2014a*; *Petry et al., 2013*). Saturation is inevitable since the building blocks of microtubules are present at a fixed concentration. In our model, we impose a carrying capacity by expressing autocatalytic nucleation as a logistic function of the local density of growing plus ends, which is qualitatively consistent with local depletion of nucleation factors such as the gamma-tubulin ring complex. Other forms of negative feedback (e.g. at the level of polymerization dynamics) are possible as well. In SI, we show that the type of negative feedback does not affect the rate of aster growth, which is determined entirely by the dynamics at the leading edge of a growing aster where the microtubule density is small and negative feedback can be neglected.

## Mathematical model of autocatalytic growth of asters

Assuming a large number of microtubules, we focus on the mean-field or deterministic dynamics (SI) and formalize our model as a set of partial differential equations. Specifically, we let $\rho_g(t,x,l)$ and $\rho_s(t,x,l)$ denote respectively the number of growing and shrinking microtubules of length $l$ with their minus ends at distance $x>0$ from the centrosome. The number of newly nucleated microtubules is given by $Q(x) = rC_g(t,x)(1 - C_g(t,x)/K)$, where $r$ is the nucleation rate, $K$ is the carrying capacity controlling the maximal plus end density, and $C_g(t,x)$ is the local density of the growing plus ends at point $x$. The polymerization dynamics and nucleation are then described by,

$$\begin{cases} \dfrac{\partial \rho_g}{\partial t} = -v_g \dfrac{\partial \rho_g}{\partial l} - f_{cat}\rho_g + f_{res}\rho_s + Q(x) \cdot \delta(l) \\ \dfrac{\partial \rho_s}{\partial t} = +v_s \dfrac{\partial \rho_s}{\partial l} + f_{cat}\rho_g - f_{res}\rho_s. \end{cases} \tag{3}$$

Note that polymerization and depolymerization changes the microtubule length $l$, but not the minus end position $x$. Equations at different $x$ are nevertheless coupled due to the nucleation term, which depends on $x$ through $C_g$.

To understand this system of equations, consider the limit of no nucleation ($r = 0$). Then, the equations at different $x$ become independent and we recover the standard model that reduces aster growth to the growth of individual microtubules (*Dogterom and Leibler, 1993*; *Verde et al., 1992*). With nucleation, aster growth is a collective phenomenon because microtubules of varying length and minus end positions contribute to $C_g(t,x)$, which can be expressed as a convolution of $\rho_g$ (see SI). The delta-function $\delta(l)$ ensures that newly nucleated microtubules have zero length.

Finally, we need to specify what happens when microtubules shrink to zero length. In our model, microtubules originating from centrosomes rapidly switch from shrinking to growth (i.e. re-nucleate), while non-centrosomal microtubules disappears completely (i.e. no re-nucleation occurs). We further assume that mother and daughter microtubules disappear without affecting each other. Indeed, if the collapse of the mother microtubule triggered the collapse of the daughter microtubule (or vice versa), then no net increase in the number of microtubules would be possible in the bounded regime. One consequence of this assumption is that the minus end of a daughter microtubule becomes detached from any other microtubules in the aster following the collapse of the mother microtubule. As a result, minus ends need to be stabilized after nucleation possibly by some additional factors (*Akhmanova and Hoogenraad, 2015*) and mechanical integrity of the aster should rely on microtubule bundling (*Ishihara et al., 2014a*).

## Asters can grow as spatially propagating waves with constant bulk density

To check if our model can describe aster growth, we solved *Equation (3)* numerically using finite difference methods in an 1D planar geometry. With relatively low nucleation rates and $J<0$, microtubule populations reached a steady-state profile confined near the centrosome reminiscent of an aster in the standard model with bounded microtubule dynamics (*Figure 2A* left). When the nucleation rate was increased, the microtubule populations expanded as a traveling wave with an approximately invariant shape and constant microtubule density at the periphery (*Figure 2A* right) consistent with the growth of interphase asters in our reconstitution experiments (*Figure 2B* and *Ishihara et al., 2014a*). Thus, our model predicted two qualitatively different states: stationary and growing asters.

## Analytical solution for aster velocity and critical nucleation

Next, we solved *Equation (3)* exactly and obtained the following analytical expression for the aster velocity in terms of model parameters:

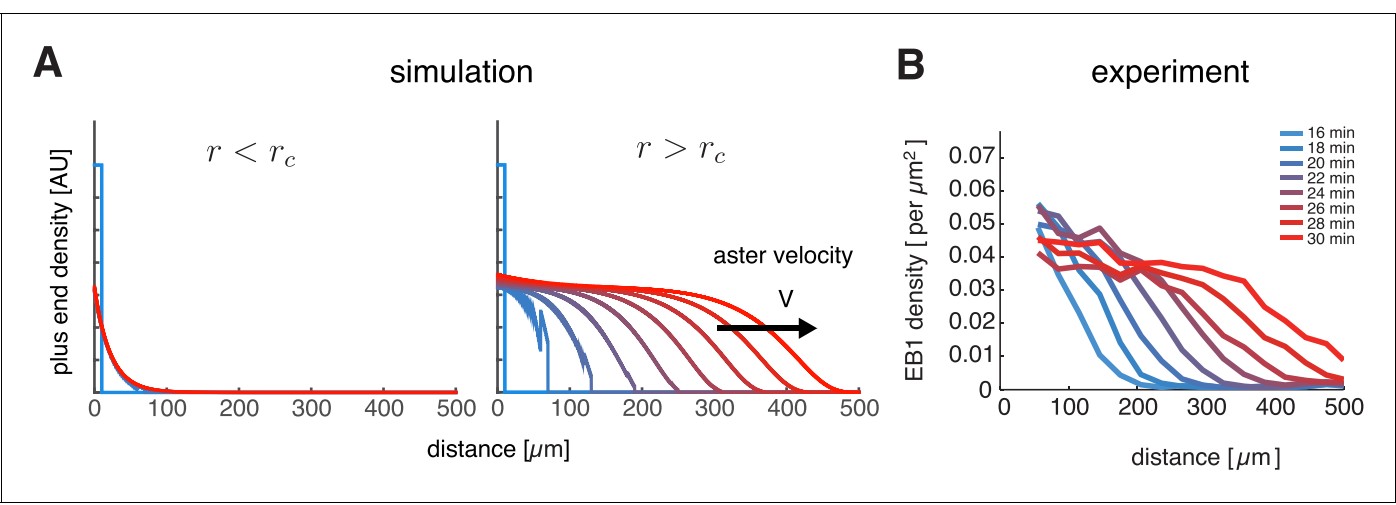

**Figure 2.** Our model captures key features of large aster growth. (A) Time evolution of growing plus end density predicted by our model, which we solved via numerical simulations in 1D geometry. In the stationary regime, the microtubule population remained near the centrosome $v_g = 30$, $v_s = 40$, $f_{cat} = 3$, $f_{res} = 1$, and $r = 1.0$ (left). In contrast, outward expansion of the microtubule population was observed when the nucleation rate was increased to $r = 2.5$, above the critical nucleation rate $r_c$ (right). For both simulations, microtubules are in the bounded regime $J<0$. (B) Experimental measurements confirm that asters expand at a constant rate with time-invariant profiles of the plus end density, as predicted by our model. The plus end densities were estimated as EB1 comet density during aster growth as previously described (*Ishihara et al., 2014a*).
Panel B reprinted with permission from Figure 4C from (Ishihara et al., 2014a), Proceedings of the National Academy of Sciences of the United States of America. Not covered by the terms of the Creative Commons Attribution 4.0 International license (© copyright Proceedings of the National Academy of Sciences of the United States of America, 2014. All Rights Reserved).

$$V = \frac{v_g(v_g f_{res} - v_s f_{cat})^2}{\left( \begin{array}{c} v_g(v_g f_{res} - v_s f_{cat})(f_{res} + f_{cat}) + (v_g + v_s)(v_g f_{res} + v_s f_{cat})r \\ -2(v_g + v_s)\sqrt{v_g f_{cat} f_{res} r(v_g f_{res} - v_s f_{cat} + v_s r)} \end{array} \right)}, \tag{4}$$

which holds for the parameter range $r_c < r < f_{cat}$. The details of the calculation, including the definition of $r_c$ are summarized in SI.

Using this expression, we summarize how aster velocity $V$ is affected by the mean polymerization rate $J$ (**Figure 3A**) and nucleation rate $r$ (**Figure 3B**). In the absence of autocatalytic nucleation ($r = 0$), our model reduces to the standard model and predicts that asters only grow when $J > 0$ with $V = J$ (**Figure 3A** blue line). When nucleation is allowed ($r > 0$), the aster velocity increases with $r$ and asters can grow even when individual microtubules are unstable $J < 0$ (**Figure 3A and B**). During this collective growth, the aster expands radially because more microtubules are nucleated than lost at the front. In the aster bulk, nucleation is reduced from the carrying capacity, and the aster exists in the dynamic balance between microtubule gain due to nucleation and loss due to depolymerization. Since microtubules are in the bounded regime, their lifetime is short, and they disappear before reaching an appreciable length. In sharp contrast to the standard model, we predict that asters are a dynamic network of short microtubules with properties independent from the distance to the centrosome. Thus, nucleation not only increases the number of microtubules, but also controls the growth rate and spatial organization of asters enabling them to span length scales far exceeding the length of an individual microtubule.

When $J < 0$, a critical nucleation rate is required for aster growth (**Figure 3B**). Indeed, microtubules constantly disappear as their length shrinks to zero, and the nucleation of new microtubules needs to occur frequently enough to overcome the microtubule loss. Consistent with this argument, our analytical solution predicts no aster growth below a certain value of nucleation (SI), termed critical nucleation rate $r_c$:

$$r_c = f_{cat} - \frac{v_g}{v_s} f_{res}. \tag{5}$$

The right hand side of this equation is the inverse of the average time that a microtubule spends in the growing state before shrinking to zero-length and disappearing (SI). Thus, aster growth requires, on average, a microtubule to nucleate at least one new microtubule during its lifetime.

The dependence of the critical nucleation rate on model parameters is very intuitive. Increasing the parameters in favor of polymerization ($v_g$ and $f_{res}$), lowers the threshold level of nucleation required for aster growth, while increasing the parameters in favor of depolymerization ($v_s$ and $f_{cat}$) has the opposite effect. We also find that $r_c = 0$ when $J = 0$, suggesting that there is no critical nucleation rate for $J \geq 0$. This limit is consistent with the standard model with $J > 0$ and $r = 0$ where the aster radius increases albeit with radial dilution of microtubule density (**Figure 1B**). The critical nucleation rate conveys the main implication of our theory: the balance between polymerization dynamics and autocatalytic nucleation defines the quantitative condition for continuous aster growth.

## Explosive transition to growth with a 'gap velocity'

At the critical nucleation rate $r = r_c$, the aster velocity $V$ takes a positive, nonzero value (**Figure 3**), which we refer to as the 'gap velocity' (SI):

$$V_{gap} \equiv \lim_{r \to r_c} V = \frac{-v_g v_s(v_g f_{res} - v_s f_{cat})}{v_g^2 f_{res} + v_s^2 f_{cat}}. \tag{6}$$

This finite jump in the aster velocity is a consequence of microtubules with finite length undergoing dynamic instability and is in sharp contrast to the behavior of reaction-diffusion systems, where traveling fronts typically become infinitesimally slow before ceasing to propagate (**Chang and Ferrell, 2013**; **Hallatschek and Korolev, 2009**; **Méndez et al., 2007**; **van Saarloos, 2003**). One can understand the origin of $V_{gap} > 0$ when microtubules are eliminated after a catastrophe event ($f_{res} = 0, J = -v_s$). In this limit, plus ends always expand with the velocity $v_g$ until they eventually collapse. Below $r_c$, this forward expansion of plus ends fails to produce aster growth because the

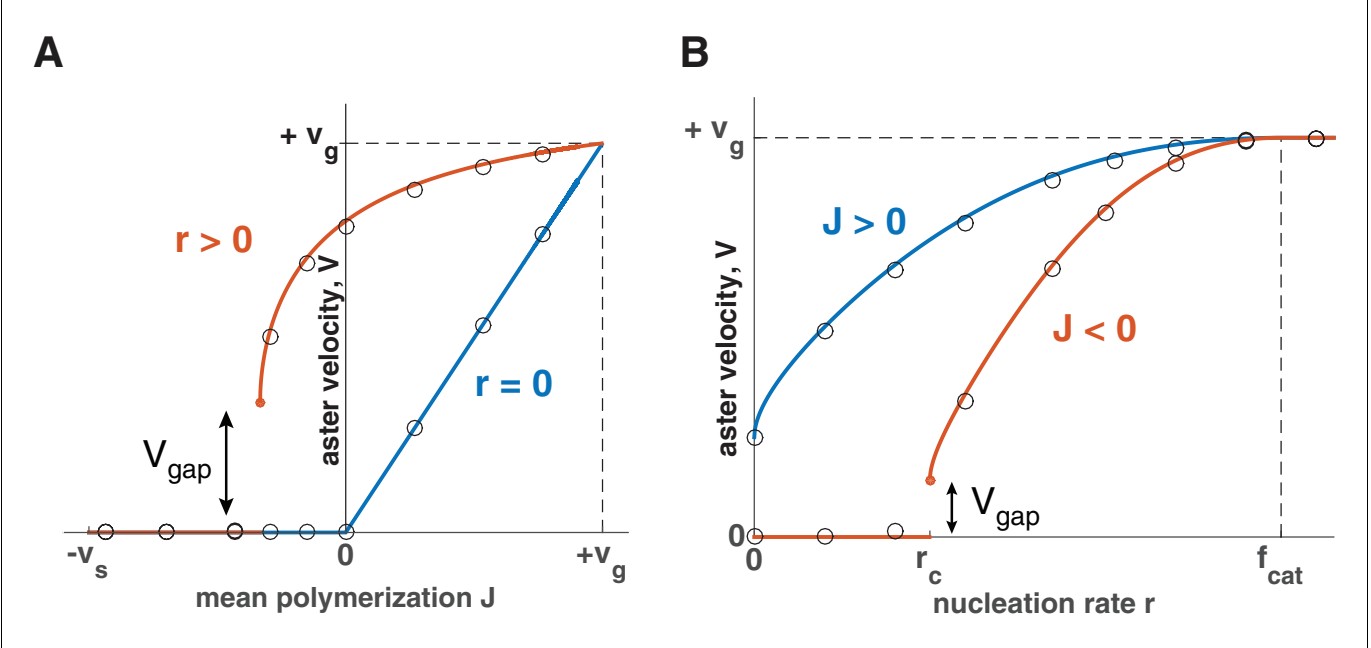

**Figure 3.** Explosive transition from stationary to growing asters and other theoretical predictions. Analytical solution (lines) and numerical simulations (dots) predict that asters either remain stationary or expand at a constant velocity, which increases with the net polymerization rate $J$ (A) and nucleation rate $r$ (B). The transition to a growing state is accompanied by a finite jump in the expansion velocity labeled as $V_{gap}$. (A) The behavior in the standard model ($r = 0$) is shown in blue and our model ($r = 1.5$) in red. Note that aster growth commences at $J < 0$ in the presence of nucleation and occurs at a minimal velocity $V_{gap}$. Although spatial growth can occur for both $J > 0$ and $J < 0$ the properties of the resulting asters could be very different (see SI). Here, $v_g = 30$, $v_s = 30$, $f_{cat} = 3$. (B) If $J < 0$, critical nucleation $r_c$ is required to commence aster growth. Blue line corresponds to $J > 0$ ($v_g = 30$, $v_s = 15$, $f_{cat} = 3$, $f_{res} = 3$) and red line to $J < 0$ ($v_g = 30$, $v_s = 15$, $f_{cat} = 3$, $f_{res} = 1$). See Materials and methods and SI for the details of the analytical solution and numerical simulations.

The following figure supplements are available for figure 3:

**Figure supplement 1.** Feedback regulation of catastrophe rate leads to the same explosive transition.

**Figure supplement 2.** Aster growth by polymer-stimulated nucleation leads to the same explosive transition.

number of plus ends declines on average. Right above $r_c$, the number of plus ends is stable, and the aster grows at the same velocity as every individual microtubule. Indeed, *Equation (6)* predicts that $V_{gap} = v_g$ when $f_{res} = 0$. The dynamics are similar for $f_{res} > 0$. At the transition, nucleation stabilizes a subpopulation of microtubules expanding forward, and their average velocity sets the value of $V_{gap}$. We also find that the magnitude of $V_{gap}$ is inversely proportional to the mean length of microtubules in the system (SI). Thus, the shorter the microtubules, the more explosive this transition becomes.

In the SI, we also show that microtubule density inside the aster is proportional to $r - r_c$. Thus, the density is close to zero during the transition from stationary to growing asters, but quickly increases as the nucleation rate becomes larger. As a result, cells can achieve rapid aster growth while keeping the density of the resulting microtubule network sufficiently low. The low density might be beneficial because of its mechanical properties or because it simply requires less tubulin to produce and energy to maintain. In addition, the explosive transition to growth with $V_{gap} > 0$ allows the cell to independently control the aster density and growth speed.

Model parameters other than the nucleation rate can also be tuned to transition asters from growth to no growth regimes. Similar to *Equation (5) and (6)*, one can define the critical parameter value and gap velocity to encompass all such transitions (*Appendix 4—table 1*). In all cases, we find that the onset of aster growth is accompanied by a discontinuous increase in the aster velocity. The finite jump in aster velocity is similarly predicted in a wide range of alternative scenarios including (i) feedback regulation of plus end dynamics (SI and *Figure 3—figure supplement 1*) and (ii) aster

growth by microtubule polymer-stimulated nucleation (SI and *Figure 3—figure supplement 2*). In summary, the gap velocity is a general prediction of the collective behavior of microtubules that are short-lived.

## Titration of MCAK slows then arrests aster growth with evidence for a gap velocity

Based on our theory, we reasoned that it would be possible to transform a growing interphase aster to a small, stationary aster by tuning polymerization dynamics and/or nucleation via biochemical perturbations in *Xenopus* egg extract. To this end, we performed reconstitution experiments in undiluted interphase cytoplasm supplied with anti-Aurora kinase A antibody coated beads, which nucleate microtubules and initiate aster growth under the coverslip (*Field et al., 2014*; *Ishihara et al., 2014a*). We explored perturbation of various regulators for plus end dynamics and nucleation. We settled on perturbation of MCAK/KIF2C, classically characterized as the main catastrophe-promoting factor in the extract system (*Kinoshita et al., 2001*; *Walczak et al., 1996*), and imaged aster growth.

In control reactions, aster radius, visualized by the plus end marker EB1-mApple, increased at velocities of 20.3±3.1 μm/min (n = 21 asters). We saw no detectable changes to aster growth with the addition of the wild type MCAK protein. In contrast, addition of MCAK-Q710 (*Moore and Wordeman, 2004*) decreased aster velocity (*Figure 4A and B*). At concentrations of MCAK-Q710 above 320 nM, most asters had small radii with very few microtubules growing from the Aurora A beads. In our model, this behavior is consistent with any change of parameter(s) that reduces the aster velocity (*Equation 4*) and arrests growth.

At 320 nM MCAK-Q710 concentration, we observed bimodal behavior. Some asters increased in radius at moderate rates, while other asters maintained a stable size before disappearing, presumably due to the decrease of centrosomal nucleation over time (*Figure 4—figure supplement 1* and *Ishihara et al., 2014a*). In particular, we observed no asters growing at velocities between 0 and 9 μm/min (*Figure 4B* and *Figure 4—figure supplement 1*). This gap in the range of possible velocities is consistent with the theoretical prediction that growing asters expand above a minimal rate $V_{gap}$.

To confirm that the failure of aster growth at high concentrations of MCAK-Q710 is caused by the changes in aster growth rather than nucleation from the beads, we repeated the experiments with *Tetrahymena* pellicles as the initiating centers instead of Aurora A beads. Pellicles are pre-loaded with a high density of microtubule nucleating sites, and are capable of assembling large interphase asters (*Ishihara et al., 2014a*). We found pellicle initiated asters to exhibit a similar critical concentration of MCAK-Q710 compared to Aurora A bead asters. While the majority of Aurora A beads subjected to the highest concentration of MCAK-Q710 lost growing microtubules over time, a significant number of microtubules persisted on pellicles even after 60 min (*Figure 4—figure supplement 2*). The radii of these asters did not change, consistent with our prediction of stationary asters. Thus, the pellicle experiments confirmed our main experimental result of small, stationary asters and that the nature of transition is consistent with the existence of a gap velocity.

Finally, we asked which parameters in our model were altered in the MCAK-Q710 perturbation. To this end, we measured the polymerization and catastrophe rates in interphase asters assembled by Aurora A beads at various MCAK-Q710 concentrations. We imaged EB1 comets at high spatio-temporal resolution, and analyzed their trajectories by tracking-based image analysis (*Applegate et al., 2011*; *Matov et al., 2010*, Materials and methods). Neither the polymerization nor the catastrophe rate changed at the MCAK-Q710 concentrations corresponding to the transition between growing and stationary asters (*Figure 4—figure supplement 3*). MCAK-Q710 has been reported to reduce microtubule polymer levels in cells (*Moore and Wordeman, 2004*), but its precise effect on polymerization dynamics and/or nucleation remains unknown. Our data are consistent with the following three scenarios for how MCAK-Q710 antagonizes microtubule assembly: (i) increased depolymerization rate, (ii) decreased rescue rate, and/or (iii) decreased nucleation rate.

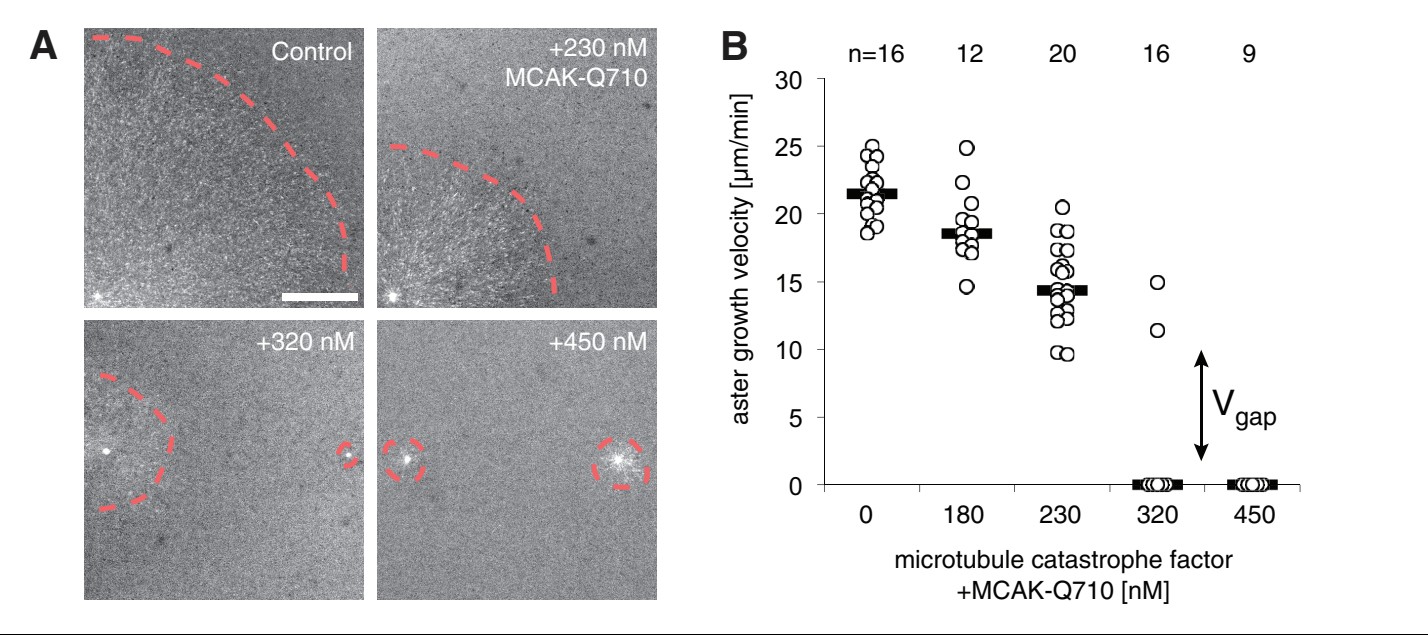

**Figure 4.** Titration of MCAK-Q710 slows then arrests aster growth through a discontinuous transition. (**A**) Addition of MCAK-Q710 results in smaller interphase asters assembled by Aurora A beads in *Xenopus* egg extract. Images were obtained 20 min post initiation with the plus end marker EB1-mApple. Dotted lines indicate the approximate outline of asters. (**B**) Aster velocity decreases with MCAK-Q710 concentration and then abruptly vanishes as predicted by the model. Note a clear gap in the values of the observed velocities and bimodality near the transition, which support the existence of $V_{gap}$. Quantification methods are described in methods and *Figure 4—figure supplement 1*.

The following figure supplements are available for figure 4:

**Figure supplement 1.** Aurora A kinase bead asters at different MCAK-Q710 concentrations.

**Figure supplement 2.** Pellicle asters at different MCAK-Q710 concentrations.

**Figure supplement 3.** Plus end polymerization rate and catastrophe rate do not significantly change with MCAK-Q710 titration.

## Discussion

### An autocatalytic model of aster growth

It has not been clear whether the standard model of aster growth can explain the morphology of asters observed in all animal cells, including those of extreme size (*Mitchison et al., 2015*). To resolve this question, we constructed a biophysical framework that incorporates microtubule polymerization dynamics and autocatalytic nucleation. Numerical simulations and analytical solutions (*Figures 2* and *3*, and *Figure 3—figure supplements 1* and *2*) recapitulated both stationary and continuously growing asters in a parameter-dependent manner. Interestingly, the explosive transition from 'growth' to 'no growth' was predicted to involve a finite aster velocity, which we confirmed in biochemical experiments (*Figure 4*).

Our biophysical model offers a biologically appealing explanation to aster growth and allows us to estimate parameters that are not directly accessible: the rescue and autocatalytic nucleation rates. For example, if we assume that MCAK-Q710 decreases the nucleation rate, we may use the $V_{gap}$ equation for $r \rightarrow r_c$ (*Equation (6)*), the equation for aster velocity V (*Equation (4)*), and our measurements of $v_g$, $v_s$, $f_{cat}$, V, and $V_{gap}$ (*Table 1*) to simultaneously estimate $f_{res}$ and r. These results are summarized in *Table 1*. Our inferred value of autocatalytic nucleation $r = 2.1$ min$^{-1}$ is comparable to previous estimates: 1.5 min$^{-1}$ (*Clausen and Ribbeck, 2007*) and 1 min$^{-1}$ (*Petry et al., 2013*) in meiotic egg extract supplemented with RanGTP. In the alternative scenarios, where MCAK-Q710 decreases the catastrophe rate or increases the depolymerization rate, our estimates of r and $f_{res}$ are

essentially the same (*Appendix 8—table 1*). Thus, our model recapitulates aster growth with reasonable parameter values and offers a new understanding for how asters grow to span large cytoplasms even when individual microtubules are unstable.

To date, few studies have rigorously compared the mechanistic consequences of plus-end-stimulated vs. polymer-stimulated nucleation. Above, we presented the theoretical predictions for aster growth by plus-end stimulated nucleation. In the SI, we also provide the results for polymer-stimulated nucleation including the critical nucleation rate *Equation A59*. Both models of nucleation have qualitatively similar behavior including the gap velocity and recapitulate experimental observations of asters growing as traveling waves. Thus, in our case, the qualitative conclusions do not depend on the precise molecular mechanism of autocatalytic nucleation. In particular, the explosive transition characterized by the gap velocity is a general prediction of modeling microtubules as self-amplifying elements whose lifetime depends on their length.

By carefully defining and quantifying autocatalytic nucleation, future studies may be able to distinguish its precise mechanism. With plus-end-stimulated nucleation, the nucleation rate $r$ has units of min$^{-1}$ and describes the number of new microtubules generated per existing plus end per minute. With polymer-stimulated nucleation, the nucleation rate has units of µm$^{-1}$ min$^{-1}$, and describes the number of new microtubules generated per micron of existing microtubule per minute. This difference has important implications for the structural mechanism of microtubule nucleation and for the prediction of cell-scale phenomena. For the issue of large aster growth, we propose specific experiments that might be able distinguish these scenarios (SI).

## Phase diagram for aster growth

How do large cells control aster size during rapid divisions? We summarize our theoretical findings with a phase diagram for aster growth in *Figure 5*. Small mitotic asters are represented by stationary asters found in the regime of bounded polymerization dynamics $J<0$ and low nucleation rates. These model parameters must change as cells transition from mitosis to interphase to produce large growing asters. Polymerization dynamics becomes more favorable to elongation during interphase (*Belmont et al., 1990*; *Verde et al., 1992*). This may be accompanied by an increased autocatalytic nucleation of microtubules.

According to the standard model, increasing $J$ to a positive value with no nucleation leads to asters in the 'individual growth' regime. A previous study suggested the interphase cytoplasm is in the unbounded polymerization dynamics $J>0$ (*Verde et al., 1992*), but our measurements of parameters used to calculate $J$ differ greatly (*Table 1*). The individual growth regime is also inconsistent with the steady-state density of microtubules at the periphery of large asters in both fish and frog embryos (*Ishihara et al., 2014a*; *Wühr et al., 2008*, *2010*). Experiments in egg extracts further confirm the addition of new microtubules during aster growth (*Ishihara et al., 2014a*) contrary to the predictions of the standard model. Furthermore, the presence of a high density of growing plus ends in the interior of growing asters in egg extract suggests that microtubules must be short compared to aster radius, and $J$ must be negative, at least in the aster interior (*Ishihara et al., 2014a*).

By constructing a model that incorporates autocatalytic nucleation $r>0$, we discovered a new regime, in which continuous aster growth is supported even when microtubules are unstable ($J<0$). We call this the 'collective growth' regime because individual microtubules are much shorter (estimated mean length of 16 µm ± 2 µm, *Table 1*) than the aster radius (hundreds of microns). Predictions of this model are fully confirmed by the biochemical perturbation via MCAK-Q710. The finite jump in the aster velocity (*Figure 4*) is in sharp contrast to the prediction of the standard models of spatial growth (*Fisher, 1937*; *Kolmogorov and Petrovskii, 1937*; *Skellam, 1951*; *van Saarloos, 2003*). Spatial growth is typically modeled by reaction-diffusion processes that account for birth events and random motion, which, in the context of microtubules, correspond to the nucleation and dynamic instability of plus ends. Reaction-diffusion models, however, neglect internal dynamics of the agents such as the length of a microtubule. As a result, such models inevitably predict a continuous, gradual increase in the aster velocity as the model parameters are varied (*Chang and Ferrell, 2013*; *Hallatschek and Korolev, 2009*; *Méndez et al., 2007*; *van Saarloos, 2003*). The observation of finite velocity jump provides a strong support for our model and rules out a very wide class of models that reproduce the overall phenomenology of aster growth including the constant velocity and profile shape (*Figure 2*). In particular, the observation of $V_{gap}$ excludes the model that we

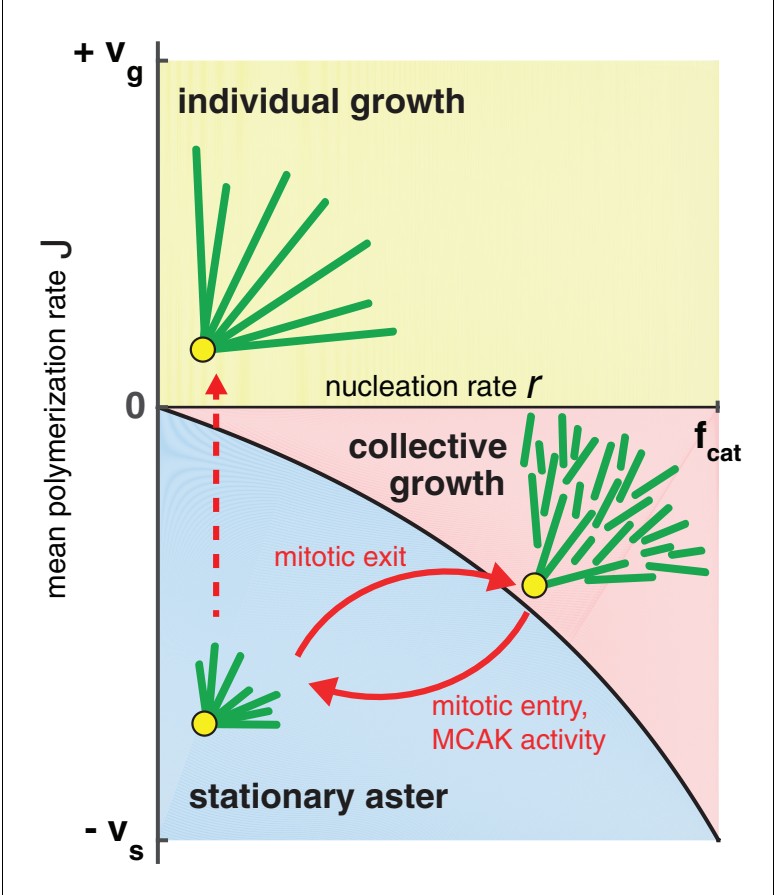

**Figure 5.** Phase diagram for aster growth. Aster morphology is determined by the balance of polymerization dynamics and autocatalytic nucleation. Small, stationary asters ($V = 0$), as observed during mitosis, occur at low nucleation $r$ and net depolymerization of individual microtubules ($J<0$). Net polymerization ($J>0$) without nucleation ($r = 0$) produces asters that expand at rate $V = J$ with dilution of microtubule density at the periphery and are thus inconsistent with experimental observations. The addition of nucleation to the individual growth regime changes these dynamics only marginally (yellow region); see SI. Alternatively, the transition from stationary to growing asters can be achieved by increasing the nucleation rate, $r$, while keeping $J$ negative. Above the critical nucleation rate $r_c$ starts the regime of collective growth ($V$ as in **Equation (4)**, which is valid for $r<f_{cat}$) that produces asters composed of relatively short microtubules (red region). The transition from stationary aster to collective growth may be achieved by crossing the curve at any location, but always involves an explosive jump in aster velocity, $V_{gap}$. The reverse transition recapitulates the results of our experimental perturbation of MCAK activity (**Figure 4**) and mitotic entry (solid arrows). We propose this unified biophysical picture as an explanation for the cell cycle dependent changes of aster morphology *in vivo*.

previously proposed based on the analogy of aster growth and the Fisher-Kolmogorov equation (**Ishihara et al., 2014b**). The implications of $V_{gap}$ for model selection are further discussed in SI.

## Collective growth of cytoskeletal structures

Our theory allows for independent regulation of aster growth rate and microtubule density through the control of the nucleation rate and microtubule polymerization. Thus, cells have a lot of flexibility in optimizing aster properties and behavior. The existence of a gap velocity results in switch-like transition from quiescence to rapid growth and allows cells to drastically alter aster morphology with a small change of parameters. Importantly, the rapid growth does not require high microtubule density inside asters, which can be tuned independently.

Collective growth produces a meshwork of short microtubules with potentially desirable properties. First, the network is robust to microtubule severing or the spontaneous detachment from the

centrosome. Second, the network can span arbitrarily large distances yet disassemble rapidly upon mitotic entry. Third, the structure, and therefore the mechanical properties, of the network do not depend on the distance from the centrosome. As a speculation, the physical interconnection of the microtubules may facilitate the transduction of mechanical forces across the cell in a way unattainable in the radial array predicted by the standard model (*Tanimoto et al., 2016*; *Wühr et al., 2010*).

The regime of collective growth parallels the assembly of other large cellular structures from short, interacting filaments (*Pollard and Borisy, 2003*) and is particularly reminiscent of how meiosis-II spindles self-assemble (*Burbank et al., 2007*; *Brugués et al., 2012*; *Brugués and Needleman, 2014*). Due to such dynamic architecture, spindles are known to have unique physical properties such as self-repair, fusion (*Gatlin et al., 2009*) and scaling (*Good et al., 2013*; *Hazel et al., 2013*; *Wühr et al., 2008*), which could allow for greater robustness and evolvability (*Kirschner and Gerhart, 1998*). Perhaps, collective growth is one of the most reliable ways for a cell to assemble cytoskeletal structures that exceed the typical length scales of individual filaments.

## Materials and methods

### Numerical simulations

We implemented a finite difference method with fixed time steps to numerically solve the continuum model (*Equation 3*). The forward Euler's discretization scheme was used except exact solutions of advection equations was used to account for the gradient terms. Specifically, the plus end positions were simply shifted by $+v_g\delta t$ for growing microtubules and by $-v_s\delta t$ for shrinking microtubules. Nucleation added new growing microtubules of zero length at a position-dependent rate given by $Q(x)$. The algorithm was implemented using MATLAB (Mathworks).

### Analytical solution

We linearized *Equation 3* for small $C_g$ and solved it using Laplace transforms in both space and time. The inverse Laplace transform was evaluated using the saddle point method (*Bender and Orszag, 1999*). We found the aster velocity as in *Equation 4*. The details of this calculation are summarized in the Supporting Text (SI).

### Aster velocity measurements

Interphase microtubule asters were reconstituted in *Xenopus* egg extract as described previously with use of p150-CC1 to inhibit dynein mediated microtubule sliding (*Field et al., 2014*; *Ishihara et al., 2014a*). Fluorescence microscopy was performed on a Nikon 90i upright microscope equipped with a Prior Proscan II motorized stage. EB1-mApple was imaged every 2 min with a 10x Plan Apo 0.45 N.A. or a 20x Plan Apo 0.75 N.A. objective. For the analysis of the aster growth front, a linear region originating from the center of asters was chosen (*Figure 4—figure supplement 1*). A low pass filter was applied to the fluorescence intensity profile and the half-max position, corresponding to the aster edge, was determined manually. The analysis was assisted by scripts written in ImageJ and MATLAB (Mathworks). Univariate scatter plots were generated with a template from (*Weissgerber et al., 2015*). EB1-mApple were purified as in (*Petry et al., 2011*), used at a final concentration of 100 nM. In some experiments, MCAK or MCAK-Q710-GFP (*Moore and Wordeman, 2004*) proteins were added to the reactions. Protein A Dynabeads coated with anti-Aurora kinase A antibody (*Tsai and Zheng, 2005*) or *Tetrahymena* pellicles were used as microtubule nucleating sites.

### Catastrophe rate measurements

Interphase asters were assembled as described above. Catastrophe rates and plus end polymerization rates were estimated from time lapse images of EB1 comets that localize to growing plus ends (*Matov et al., 2010*). The distributions of EB1 track durations were fitted to an exponential function to estimate the catastrophe rate. Spinning disc confocal microscopy was performed on a Nikon Ti motorized inverted microscope equipped with Perfect Focus, a Prior Proscan II motorized stage, Yokagawa CSU-X1 spinning disk confocal with Spectral Applied Research Aurora Borealis modification, Spectral Applied Research LMM-5 laser merge module with AOTF controlled solid state lasers: 488 nm (100 mW), 561 nm (100 mW), and Hamamatsu ORCA-AG cooled CCD camera. EB1-mApple

was imaged every 2 s with a 60x Plan Apo 1.40 N.A. objective with 2×2 binning. EB1 tracks were analyzed with PlusTipTracker (*Applegate et al., 2011*).

## Video abstract

A 2 min video abstract of this paper is available at https://youtu.be/jfjA2S-fE9U.

## Acknowledgements

We thank the members of Mitchison and Korolev groups for helpful discussion. This work was supported by NIH grant GM39565 and by MBL summer fellowships. The computations in this paper were run on the Odyssey cluster supported by the FAS Division of Science, Research Computing Group at Harvard University. We thank Nikon Imaging Center at Harvard Medical School, and Nikon Inc. at Marine Biological Laboratory for microscopy support. We thank Ryoma Ohi for providing MCAK and expert advice. We thank Linda Wordeman for providing MCAK-Q710-GFP and expert advice. KK was supported by a start up fund from Boston University and by a grant from the Simons Foundation #409704. KI was supported by the Honjo International Scholarship Foundation.

## Additional information

### Funding

| Funder | Grant reference number | Author |
| --- | --- | --- |
| National Institutes of Health | GM39565 | Keisuke Ishihara<br>Timothy J Mitchison |
| Marine Biological Laboratory | Whitman Center Award | Timothy J Mitchison |
| Honjo International Scholarship Foundation | Graduate Student Fellowship | Keisuke Ishihara |
| Boston University | Start up fund | Kirill S Korolev |
| Simons Foundation | #409704 | Kirill S Korolev |

The funders had no role in study design, data collection and interpretation, or the decision to submit the work for publication.

### Author contributions

KI, Conception and design, Development and analysis of the model, Acquisition of data, Analysis and interpretation of data, Drafting and revising the article; KSK, Conception and design, Development and analysis of the model, Analysis and interpretation of data, Drafting and revising the article; TJM, Conception and design, Analysis and interpretation of data, Drafting and revising the article

### Author ORCIDs

Keisuke Ishihara, http://orcid.org/0000-0002-8481-8680
Timothy J Mitchison, http://orcid.org/0000-0001-7781-1897

### Ethics

Animal experimentation: This study was performed in strict accordance with the recommendations in the Guide for the Care and Use of Laboratory Animals of the National Institutes of Health. All of the animals were handled according to approved institutional animal care and use committee (IACUC) protocols (IS00000519) of Harvard Medical School. The protocol was approved by the HMA Standing Committee on Animals (SIRIUS/Procurement Number 2762).

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

## Appendix 1

### A continuum model of aster growth

To describe our system of microtubules, we define the mean-field variable $\rho_g(t,x,l)$ which represents the local density of growing plus ends of length $l$ with its corresponding minus end at position $x$ at time $t$. Similarly, we define the shrinking plus end density $\rho_s(t,x,l)$. We assume that all microtubules have the same polarity, namely that all microtubules have their plus ends pointing outwards. Given this polarity, the plus end position of a microtubule is $x+l$. We define nucleation as the birth of zero-length growing microtubules. The nucleation rate is denoted by $Q(t,x)$. The evolution of our system is described as follows:

$$\begin{cases} \dfrac{\partial \rho_g}{\partial t} = -v_g \dfrac{\partial \rho_g}{\partial l} - f_{cat}\rho_g + f_{res}\rho_s + Q(t,x)\cdot\delta(l), \\ \dfrac{\partial \rho_s}{\partial t} = +v_s \dfrac{\partial \rho_s}{\partial l} + f_{cat}\rho_g - f_{res}\rho_s, \end{cases} \tag{A1}$$

where $\delta(l)$ is the Dirac delta function. (Instead of a delta function as in **Equation (A1)**, one can introduce nucleation as a boundary condition specifying the flux of new microtubules into the system, $Q(t,x) = v_g\rho_g(t,x,l=0)$.) The above expression represents an infinite set of equations valid for the continuum $x \geq 0$.

In general, the nucleation rate may depend on $\rho_g$ and $\rho_s$. In the following sections, we assume that nucleation proceeds as bifurcation of growing plus ends.

Nucleation is expressed as a logistic function of local growing plus end density with carrying capacity of the system being $K$. (An alternative mechanism for autocatalytic nucleation is a scenario where the local polymer density stimulates nucleation. This may better relate to proposed models in which freely diffusing nucleation complexes bind to the side of pre-existing microtubules and become activated. The nucleation term in **Equation (A2)** has the advantage that the exact asymptotic solution may be obtained and we argue that it captures the qualitative effect of autocatalytic nucleation.)

$$Q(t,x) = v_g\,\rho_g(t,x,l=0) = r \cdot C_g(t,x)\left(1 - \frac{C_g(t,x)}{K}\right), \tag{A2}$$

where $C_g(t,x)$ denotes the local growing plus end density at position $x$ and time $t$. By equating the plus end density times the surface area of a sphere of radius $x$ to the integration of $\rho_g(t,x,l)$ whose plus ends happens be positioned at distance $x$, we obtain the following expression for local growing plus end density:

$$C_g(t,x) = \int_0^{+\infty} dl \int_0^x dx_- \, \rho_g(t,x_-,l)\cdot\delta(x-x_--l) \tag{A3}$$

Similarly, we define the shrinking plus end density $C_s(t,x_+)$.

In higher dimensions, **Equation (A3)** generalizes as follows

$$C_g(t,|x|) = \frac{1}{|x|^{d-1}} \int_0^{+\infty} dl \int_0^{|x|} d|x_-| \, \rho_g(t,|x_-|,l)\cdot\delta(|x|-|x_-|-l)\,|x|^{d-1}, \tag{A4}$$

where we have assumed spherical geometry.

## Appendix 2

### Bounded and unbounded regimes of polymerization dynamics

In the absence of nucleation ($r = 0$), the dynamics of our system is goverened solely by microtubule plus end dynamics. One distinguishes two types of growth: bounded and unbounded. In the former case, the mean polymerization rate $J = \frac{v_g f_{res} - v_s f_{cat}}{f_{res} + f_{cat}}$ is negative and microtubules shrink on average. In the latter case, $J$ is positive and microtubule become progressively longer. $J$ is essentially the directional bias of the plus end dynamics, the drift term of the biased random walk (**Bicout, 1997**; **Dogterom and Leibler, 1993**; **Verde et al., 1992**).

Imagine a scenario where there is a fixed number of microtubules in the system, and that all their minus ends are at the origin. Further, let us assume that when plus ends shrink back to their minus ends, they instantly transition to a growing state, akin to a reflective boundary condition at the origin. When $J<0$, the system will reach a steady-state where the length of individual microtubules are found to be exponentially distributed with an average length of $\langle l \rangle = \frac{-v_g v_s}{v_g f_{res} - v_s f_{cat}}$. When $J>0$, eventually all microtubules will be long enough with their plus ends far from the origin. Thus, there is no steady-state length distribution and the average length increases at rate $J$.

## Appendix 3

### Aster growth dynamics with autocatalytic microtubule nucleation

Here, we analyze the behavior of the system when polymerization dynamics and autocatalytic microtubule nucleation are allowed. Our intuition is that microtubule nucleation will produce microtubules at distances far from the origin, and that with high enough nucleation, the population of microtubules will start to move away from the origin as a self-propagating wave. Waves with growth terms that monotonically decline with density are called pulled fronts, and can be analyzed through linearization (*van Saarloos, 2003*). This is the case for the microtubule nucleation specified by *Equation (A2)*, and the full analytical solution for the spatiotemporal evolution can be obtained. In particular, we find the asymptotic front velocity of microtubules as a function of model parameters. This leads us to the concept of the critical nucleation rate, which summarizes the condition that determines aster growth (linear increase in radius) vs. no growth (constant radius). Other feedback mechanisms in $r, f_{cat}, v_g,$ *etc.* lead to the same final equations where all parameters are specified at low density (see Appendix 7 for an extended discussion).

### Solution of the system

We apply the Laplace transform to *Equation (A1)* in the following way: time domain $t \rightarrow s$, spatial domain $x \rightarrow k$ and length domain $l \rightarrow q$. Our system recast in $\rho_g(s,k,q)$ and $\rho_s(s,k,q)$ becomes

$$\begin{cases} s\rho_g - \rho_g(t=0,k,q) &= -v_g q\rho_g + v_g \rho_g(s,k,l=0) - f_{cat}\rho_g + f_{res}\rho_s, \\ s\rho_s - \rho_s(t=0,k,q) &= +v_s q\rho_s - v_s \rho_s(s,k,l=0) + f_{cat}\rho_g - f_{res}\rho_s. \end{cases} \quad (A5)$$

We can not directly solve the system *Equation (A5)* for $\rho_g(s,k,q)$ and $\rho_s(s,k,q)$ as $\rho_g(s,k,l=0)$ and $\rho_s(s,k,l=0)$ are unknown. However, we demonstrate that the system can be closed for the local plus end densities $C_g(s,k)$ and $C_s(s,k)$ when the nucleation rate depends on $\rho_g(t,k,l)$ and $\rho_s(t,k,l)$ as in *Equation (A2)*. We substitute this solution back to *Equation (A5)* and obtain the full solution of the system in terms of $\rho_g$ and $\rho_s$.

First, let us consider the transformation of $C_g(t,x)$ in the spatial domain $x \rightarrow k$.

$$\begin{aligned} C_g(t,k) &= \int_0^{+\infty} e^{-kx} dx\, C_g(t,x) \\ &= \int_0^{+\infty} e^{-kx} dx \int_0^{+\infty} dl \int_0^{+\infty} dx\, \delta(x-x_- - l)\,\rho_g(t,x_-,l) \\ &= \int_0^{+\infty} \int_0^{+\infty} dx\, dl\, \rho_g(t,x,l)\, e^{-kx} e^{-kl} \\ &= \int_0^{+\infty} dl\, \rho_g(t,k,l)\, e^{-kl} \\ &= \rho_g(t,k,q=k) \end{aligned} \quad (A6)$$

We apply the same transform to $C_s(t,x)$ and obtain the following:

$$C_g(s,k) = \rho_g(s,k,k) \quad \text{and} \quad C_s(s,k) = \rho_s(s,k,k) \quad (A7)$$

Thus, we find that the local plus end density $C_g(s,k)$ is equivalent to a special subset of $\rho_g(s,k,q)$ that is $\rho_g(s,k,k)$, where the spatial and length domains are coupled. By applying **Equation (A7)** and $k=q$ to **Equation (A5)**, we obtain

$$\begin{cases} sC_g - C_g(t=0,k) &= -v_g k C_g + v_g \rho_g(l=0) - f_{cat}C_g + f_{res}C_s, \\ sC_s - C_s(t=0,k) &= +v_s k C_s - v_s \rho_s(l=0) + f_{cat}C_g - f_{res}C_s. \end{cases} \tag{A8}$$

Note that the term corresponding to growing plus ends at $l=0$ is equivalent to our definition of nucleation rate **Equation (A2)**. Here, we substitute the linearized form of the boundary condition with **Equation (A2)**, $v_g \rho_g(s,\,k,\,l=0) = r \cdot C_g(s,k)$, which is valid for small $C_g$ or at the leading edge of the aster. The term corresponding to shrinking plus ends at $l=0$ is not directly specified in our system. The value of $v_s \rho_s(l=0)$ can be obtained from a special condition required for physical consistency as we show below. For now, we will treat $\rho_s(l=0)$ as known.

From the initial conditions $\rho_g^0 = \rho_g(t=0,k,q)$ and $\rho_s^0 = \rho_s(t=0,k,q)$, we know $C_g(t=0,k) = \rho_g(t=0,k,k)$ and $C_s(t=0,k) = \rho_s(t=0,k,k)$. We arrive at the subproblem:

$$A_r \begin{pmatrix} C_g(s,k) \\ C_s(s,k) \end{pmatrix} = \begin{pmatrix} \rho_g(t=0,k,k) \\ \rho_s(t=0,k,k) - v_s \rho_s(l=0) \end{pmatrix},$$

$$\text{where } A_r = \begin{bmatrix} s + v_g k + f_{cat} - r & -f_{res} \\ -f_{cat} & s - v_s k + f_{res} \end{bmatrix}. \tag{A9}$$

This system is solved by matrix inversion. The solution for growing plus end density is

$$C_g(s,k) = \frac{1}{\det(A_r)} \left[ A_{r,22} \rho_g(t=0,k,k) - A_{r,12}(\rho_s(t=0,k,k) - v_s \rho_s(l=0)) \right]. \tag{A10}$$

With the knowledge of $C_g(s,k)$, we proceed to find the general solution for $\rho_g(s,k,q)$ by returning to the full problem **Equation (A1)**. Arranging the known quantities to the right hand side, the rewritten problem reads

$$A \begin{pmatrix} \rho_g(s,k,q) \\ \rho_s(s,k,q) \end{pmatrix} = \begin{pmatrix} \rho_g^0 + r C_g(s,k) \\ \rho_s^0 - v_s \rho_s(l=0) \end{pmatrix},$$

$$\text{where } A = \begin{bmatrix} s + v_g q + f_{cat} & -f_{res} \\ -f_{cat} & s - v_s q + f_{res} \end{bmatrix}. \tag{A11}$$

For the solution of plus end density, we substitute our solution of $C_g(s,k)$ and obtain

$$\begin{aligned} \rho_g(s,k,q) &= \frac{1}{\det(A)} \left[ A_{22}(\rho_g^0 + r C_g(s,k)) - A_{12}(\rho_s^0 - v_s \rho_s(l=0)) \right] \\ &= \frac{1}{\det(A)} \left( A_{22}\rho_g^0 - A_{12}\rho_s^0 \right) + \frac{A_{12}}{\det(A)} v_s \rho_s(l=0) + \\ & \quad \frac{1}{\det(A)} \frac{r A_{22}}{\det(A_r)} \left( A_{r,22}\rho_g(t=0,k,k) - A_{r,12}\rho_s(t=0,k,k) + A_{r,12} v_s \rho_s(l=0) \right) \\ &= \frac{1}{\det(A)} \left( A_{22}\rho_g^0 - A_{12}\rho_s^0 \right) + v_s \rho_s(l=0) \left( \frac{A_{12}}{\det(A)} + \frac{r A_{22} A_{r,12}}{\det(A)\det(A_r)} \right) \\ & \quad + \frac{r A_{22}}{\det(A)\det(A_r)} \left( A_{r,22}\rho_g(t=0,k,k) - A_{r,12}\rho_s(t=0,k,k) \right) \end{aligned} \tag{A12}$$

## Physical plausibility at $l \to \infty$

In *Equation (A12)*, we still have one unknown $\rho_s(l=0)$, which we determine by imposing a physical plausibility condition. Namely, we impose that the number of infinitely long microtubules are zero by requesting that the solution $\rho_g$ (and $\rho_s$) decays as $l \to \infty$. To obtain $\rho_s(l=0)$, we apply the inverse Laplace transform $q \to l$,

$$
\begin{aligned}
\rho_g(s,k,l) &= \frac{1}{2\pi i} \int_{\gamma-i\infty}^{\gamma+i\infty} e^{ql} dq\, \rho_g(s,k,q) \\
&= \sum_j \operatorname*{Res}_{q_j} e^{q_j l} \rho_g(s,k,q_j)
\end{aligned}
\tag{A13}
$$

Note that the only term that depends on $q$ and can give rise to a residue in *Equation (A12)* is $\det(A)$. The condition $\det(A) = 0$ is equivalent to

$$
q^2 v_g v_s - q[s(v_g - v_s) + v_g f_{res} - v_s f_{cat}] - [s^2 + s(f_{cat} + f_{res})] = 0
\tag{A14}
$$

This equation has two roots $q_+ > 0$ and $q_- < 0$, since

$$
q_+ q_- = -\frac{s^2 + s(f_{cat} + f_{res})}{v_g v_s} < 0.
\tag{A15}
$$

Therefore, the $\rho_g(s,k,l)$ takes the form $\rho_g(s,k,l) = e^{q_+ l} \rho_g(s,k,q_+) + e^{q_- l} \rho_g(s,k,q_-)$.

We require that $\rho_g(s,k,l)$ takes a finite value in the limit of $l \to \infty$ by setting the coefficient for $e^{q_+ l}$ to zero, in other words $\rho_g(s,k,q_+) = 0$. Thus, with $A^+$ denoting matrix $A$ when $q = q_+$ and $\rho_g^{0,+} = \rho_g(t=0,k,q^+)$, we request the following condition:

$$
\begin{aligned}
& A_{22}^+ \rho_g^{0,+} - A_{12}^+ \rho_s^{0,+} + v_s \rho_s(l=0)\left(A_{12}^+ + \frac{rA_{22}^+ A_{r,12}}{\det(A)}\right) \\
& + \frac{rA_{22}^+}{\det(A)}\left(A_{r,22}\rho_g(t=0,k,k) - A_{r,12}\rho_s(t=0,k,k)\right) = 0
\end{aligned}
\tag{A16}
$$

Recall that $\rho_s(l=0)$ was introduced as an unknown in our system via Laplace transform despite the absence of a *bona fide* boundary condition for shrinking microtubules of length zero. We may now solve the above equation (Similar arguments were previously used by (*Bicout, 1997*) in the context of bounded, non-expanding, asters) for $\rho_s(l=0)$, and substitute it to *Equation (A12)*, resulting in our final solution

$$
\begin{aligned}
\rho_g(s,k,q) &= \frac{A_{22}\rho_g^0 - A_{12}\rho_s^0}{\det(A)} + \frac{rA_{22}(A_{r,22}\rho_g(t=0,k,k) - A_{r,12}\rho_s(t=0,k,k))}{\det(A)\det(A_r)} \\
&+ \frac{1}{\det(A)\det(A_r)} \cdot \frac{A_{12}\det(A_r) + rA_{22}A_{r,12}}{A_{12}^+\det(A_r) + rA_{22}^+ A_{r,12}} \cdot \left(-\rho_g^{0,+} A_{22}^+ \det(A_r)\right. \\
&\left. -\rho_g(t=0,k,k) rA_{22}^+ A_{r,22} + \rho_s^{0,+} A_{12}^+ \det(A_r) + \rho_s(t=0,k,k) rA_{22}^+ A_{r,12}\right).
\end{aligned}
\tag{A17}
$$

## Summary of solutions $C_g(s,k)$ and $\rho_g(s,k,q)$

Assuming of microtubule nucleation as in *Equations (A2) and (A3)*, we have derived the full solution for $\rho_g(s,k,q)$ written as *Equation (A17)*. The solution for $\rho_s(s,k,q)$ may be derived similarly. These solutions directly correspond to the plus end density solutions $C_g(s,k) = \rho_g(s,k,k)$ and $C_s(s,k) = \rho_s(s,k,k)$. Direct experimental measurements are available for $C_g(t,x)$.

As we are primarily interested in the velocity at which the front of plus ends advance in the long time limit, we proceed with our analysis focusing on the behavior of $C_g(t,x)$.

$C_g(t,x).$

## Appendix 4

# Critical transition to aster growth

## Dispersion relations for $C_g(t,x)$

We wish to determine the velocity at which the aster expands its radius. We apply the inverse Laplace transform $s \to t$ to $C_g(t,k)$

$$
\begin{aligned}
C_g(t,k) &= \frac{1}{2\pi i}\int_{\gamma-i\infty}^{\gamma+i\infty} e^{st}ds\, C_g(s,k) \\
&= \sum_j \mathrm{Res}\, C_g(s_j(k),k)\, e^{s_j(k)t},
\end{aligned}
\tag{A18}
$$

where the subscript $j$ specifies the different poles in $C_g(s,k)$. Further, we apply the inverse Laplace transform $k \to x$,

$$
\begin{aligned}
C_g(t,x) &= \frac{1}{2\pi i}\int_{\gamma-i\infty}^{\gamma+i\infty} e^{xk}dk\, C_g(t,k) \\
&= \frac{1}{2\pi i}\int_{\gamma-i\infty}^{\gamma+i\infty} dk \sum_j \mathrm{Res}\, C_g(s_j(k),k)\, e^{s_j(k)t}e^{xk}.
\end{aligned}
\tag{A19}
$$

Finally, we transform the spatial variable to the right moving reference frame $z = x - Vt$ where $V>0$ is the aster velocity we wish to determine,

$$
\begin{aligned}
C_g(t,z) &= \frac{1}{2\pi i}\int_{\gamma-i\infty}^{\gamma+i\infty} dk \sum_j \mathrm{Res}\, C_g(s_j(k),k)\, e^{s_j(k)t}e^{(z+Vt)k} \\
&= \frac{1}{2\pi i}\int_{\gamma-i\infty}^{\gamma+i\infty} dk \sum_j \mathrm{Res}\, C_g(s_j(k),k)\, e^{kz}e^{(s_j(k)+kV)t}.
\end{aligned}
\tag{A20}
$$

We must evaluate this integral in the long time limit $t \to \infty$ which corresponds to the solution that describes the front of the expanding aster. Note that the integral takes the form of $\int f(k)e^{tg_j(k)}dk$ where $t$ is large. We follow 'steepest descent' or 'saddlepoint method' and approximate the integral by the contribution from the saddlepoint $k^*$ for $g_j(k) = s_j(k) - kV$. We also impose a time invariance condition by requesting the real part of $g_j(k)$ to be zero at this point. In other words, the conditions that yield the asymptotic solution are:

$$
\begin{cases}
\dfrac{d(s_j(k)+kV)}{dk}\Big|_{k=k^*} = 0 \\
\mathrm{Re}(s_j(k^*)+k^*V) = 0
\end{cases}
\tag{A21}
$$

These two equations together with the equation that specifies the poles allow us to specify the pairs of $s_j$ and $k$ that describes the shape and velocity of the expanding front.

We return to *Equation (A17)* and examine how poles could arise. There are three possibilities $\det(A) = 0$, $\det(A_r) = 0$, and $A_{12}^+ + \frac{rA_{22}^+ A_{r,12}}{det(A_r)} = 0$. We examine them in order:

- $\det(A) = 0 \Leftrightarrow k = q_+$ is not a pole, since $A_{22} \to A_{22}^+$, $\rho_g(t=0,k=q_+,k=q_+) = \rho_g^{0,+}$, etc. and the relevant numerator becomes zero, i.e. $C_g(s,k)$ is not singular and there is no residue.

- $\det(A_r) = 0$ does not lead to a pole either because the second and the third term in *Equation (A17)* cancel.

- $A_{12}^+ + \frac{rA_{22}^+ A_{r,12}}{\det(A_r)} = 0 \Leftrightarrow \det(A_r) = -r(s - v_s q_+ + f_{res})$ is a pole.

Thus, the only pole of the equation **Equation (A17)**, $\det(A_r) = -r(s - v_s q_+ + f_{res})$, specifies the asymptotic, traveling front solution of our system.

## Aster velocity

The velocity of the aster growth is derived from the conditions imposed by **Equation (A21)**. It is easy to see that both $s^*$ and $k^*$ are real numbers, so we rewrite these conditions as follows:

$$V = -\frac{d\,s(k)}{dk}\Big|_{k=k_i^*} = -\frac{s^*}{k^*} \tag{A22}$$

In the proceeding section, we denote $s^*$ as $s$ and $k^*$ and $k$ for simplicity. The pole is specified by $\det(A_r) = -r(s - v_s q_+ + f_{res}) \Leftrightarrow$

$$(s + v_g k + f_{cat} - r)(s - v_s k + f_{res}) - f_{res}f_{cat} = -r(s - v_s q_+ + f_{res}) \quad \Leftrightarrow$$
$$s^2 + [(v_g - v_s)k + f_{cat} + f_{res}]s + [-v_g v_s k^2 + (v_g f_{res} - v_s f_{cat} + v_s r)k - v_s r q_+] = 0 \tag{A23}$$

Collectively, **Equations (A14), (A22), and (A23)** specifies the four unknowns, $s$, $k$, $q_+$, and $V$.

Since **Equations (A14) and (A23)** become identical when $k = q_+$, we reject this trivial case. Then, these two equations lead us to

$$k = \frac{(v_g - v_s)s + (v_g f_{res} - v_s f_{cat} + v_s r)}{v_g v_s} - q_+. \tag{A24}$$

Differentiating **Equation (A24)**, we find

$$\frac{dk}{ds} = \frac{v_g - v_s}{v_g v_s} - \frac{dq_+}{ds}. \tag{A25}$$

Dividing **Equation (A24)** by $s$, we find

$$\frac{k}{s} = \frac{v_g - v_s}{v_g v_s} + \frac{v_g f_{res} - v_s f_{cat} + v_s r}{v_g v_s} \cdot \frac{1}{s} - \frac{q_+}{s}. \tag{A26}$$

Using **Equation (A22)**, we equate **Equations (A25) and (A26)** and find,

$$\frac{dq_+}{ds} = \frac{q_+}{s} - \frac{v_g f_{res} - v_s f_{cat} + v_s r}{v_g v_s} \cdot \frac{1}{s}. \tag{A27}$$

Differentiating **Equation (A14)** by $s$, we solve for $\frac{dq_+}{ds}$ and find,

$$\frac{dq_+}{ds} = \frac{(v_g - v_s)q_+ + 2s + f_{cat} + f_{res}}{2q_+ v_g v_s + (v_g - v_s)s + v_g f_{res} - v_s f_{cat}}. \tag{A28}$$

Eliminating $\frac{dq_+}{ds}$ from **equations (A27) and (A28)**, we obtain

$$q_+ = \frac{(v_g f_{res} - v_s f_{cat} + v_s r)(v_g f_{res} - v_s f_{cat}) + s(v_g^2 f_{res} + v_s^2 f_{cat} + v_s r(v_g - v_s))}{v_g v_s (v_g f_{res} - v_s f_{cat} + 2 v_s r)}.$$

We substitute this into **Equation (A14)**, and choose the positive root for $s$.

$$
\begin{aligned}
s = \frac{1}{(f_{cat} - r)(v_g + v_s)(v_g f_{res} + v_s r)} &\cdot \Big( r(v_g f_{res} + v_s f_{cat})(v_g f_{res} - v_s f_{cat} + v_s r) \\
&+ (v_g f_{res} - v_s f_{cat} + 2 v_s r)\sqrt{v_g f_{cat} f_{res} r(v_g f_{res} - v_s f_{cat} + v_s r)} \Big)
\end{aligned}
\tag{A29}
$$

Using **Equation (A24)**, we find

$$
\begin{aligned}
k = \frac{-1}{v_g(f_{cat} - r)(v_g + v_s)(v_g f_{res} + v_s r)} &\cdot \Big( r(v_g f_{res} - v_s f_{cat} + v_s r)(v_g(f_{res} - f_{cat} + r) + v_s r) \\
&+ (v_g(f_{res} + f_{cat} - r) + v_s r)\sqrt{v_g f_{cat} f_{res} r(v_g f_{res} - v_s f_{cat} + v_s r)} \Big)
\end{aligned}
\tag{A30}
$$

$k$ determines the rate of spatial decay of the density at the front, which follows from **Equation (A20)**.

We require $r < f_{cat}$ as $k^*$ diverges at $r = f_{cat}$. The velocity of the propagating front is then given by

$$
V = -\frac{s}{k} = \frac{v_g(v_g f_{res} - v_s f_{cat})^2}{\left( \begin{aligned} &v_g(v_g f_{res} - v_s f_{cat})(f_{res} + f_{cat}) + (v_g + v_s)(v_g f_{res} + v_s f_{cat})r \\ &-2(v_g + v_s)\sqrt{v_g f_{cat} f_{res} r(v_g f_{res} - v_s f_{cat} + v_s r)} \end{aligned} \right)}.
\tag{A31}
$$

Our expression is valid for the range $r_c \leq r \leq f_{cat}$. For $r < r_c$, the aster fails to expand and reaches a steady state size with limited radius. For $r > f_{cat}$, we expect some microtubules or the microtubules they nucleated to polymerize without ever experiencing a growth to shrinkage transition. In this scenario, we expect the very periphery of the aster to expand at polymerization rate $v_g$. We discuss $r_c$ in more detail in the following section.

For the special case of $J = 0$, **Equation (A31)** has the numerator equal to zero, so we return to **Equations (A29) and (A30)** and find the aster velocity as:

$$
V = \frac{v_g(v_g f_{res} + v_s f_{cat} + 2\sqrt{v_g v_s f_{cat} f_{res}})}{v_g(f_{res} - f_{cat} + r) + v_s r + \frac{v_g(f_{res} + f_{cat} - r) + v_s r}{v_s r}\sqrt{v_g v_s f_{cat} f_{res}}}
\tag{A32}
$$

## Critical nucleation rate and gap velocity

We define the critical nucleation rate $r_c$ as the minimum value of nucleation $r$ at which the system results in front propagation. As seen in **Equation (A31)**, the aster expansion velocity takes a real value as long as the term inside the square root of $v_g f_{res} - v_s f_{cat} + v_s r$ is positive. For $r < r_c$, there is no real solution for $V$, while, for $r > r_c$, a pair of solutions exists. One of them predicts that $V$ decreases with $r$ and is therefore unphysical. **Equation (A31)** specifies other solution of this pair.

The critical nucleation rate is:

$$
r_c = f_{cat} - \frac{v_g}{v_s} f_{res}
\tag{A33}
$$

When $r = r_c$, aster expansion velocity takes a finite value, which we term the 'gap velocity'.

$$V_{gap} = \lim_{r \to r_c} V = \frac{v_g v_s (-v_g f_{res} + v_s f_{cat})}{v_g^2 f_{res} + v_s^2 f_{cat}} \qquad (A34)$$

Note that new microtubules nucleate only on growing plus ends; therefore, nucleation events preferentially occur on microtubules that are in the growing state more often than expected on average. As a result, the subpopulation of microtubules stabilized by nucleation expands at a velocity larger than that of a typical microtubule. In fact, the velocity of a typical microtubule is $J$, which is negative, while the velocity of the subpopulation of microtubules that sets $V_{gap}$ is positive.

We also find that $V_{gap}$ is inversely proportional to the mean microtubule length $\langle l \rangle$,

$$\begin{aligned} V_{gap} &= \frac{v_g v_s}{v_g^2 f_{res} + v_s^2 f_{cat}} \cdot \frac{v_g f_{res} - v f_{cat}}{-v_g v_s} \cdot v_g v_s \\ &= \frac{v_g^2 v_s^2}{v_g^2 f_{res} + v_s^2 f_{cat}} \cdot \frac{1}{\langle l \rangle}. \end{aligned} \qquad (A35)$$

This points to us that the origin of $V_{gap}$ is the finite length of microtubules in the system. The shorter the microtubules are, the more explosive the transition becomes.

In a similar manner, we can define the critical transition with respect to any of the five parameters in the system. Thus, we expand our definition of gap velocity to encompass all such limits. The gap velocities defined by the change of a single parameter are listed in **Appendix 4—table 1**.

**Appendix 4—table 1.** Gap velocities defined by different critical parameters.

| Critical parameter | $V_{gap}$ |
| --- | --- |
| $r_c = f_{cat} - \dfrac{v_g}{v_s} f_{res}$ | $\dfrac{v_g v_s (-v_g f_{res} + v_s f_{cat})}{v_g^2 f_{res} + v_s^2 f_{cat}}$ |
| $v_{g,c} = v_s \dfrac{f_{cat} - r}{f_{res}}$ | $\dfrac{r(f_{cat} - r)v_s}{f_{cat}^2 + f_{cat}(f_{res} - 2r) + r^2}$ |
| $v_{s,c} = v_g \dfrac{f_{res}}{f_{cat} - r}$ | $\dfrac{r f_{res} v_g}{f_{cat}^2 + f_{cat}(f_{res} - 2r) + r^2}$ |
| $f_{cat,c} = r + \dfrac{v_g}{v_s} f_{res}$ | $\dfrac{r v_g v_s^2}{r v_s^2 + f_{res} v_g (v_g + v_s)}$ |
| $f_{res,c} = \dfrac{v_s}{v_g}(f_{cat} - r)$ | $\dfrac{r v_g v_s}{-r v_g + f_{cat}(v_g + v_s)}$ |

## Aster growth dynamics when $J > 0$

Past the transition to the traveling wave regime, further changes in model parameters can make the mean polymerization rate $J$ positive. At this point, aster velocity shows no unexpected behavior and changes smoothly as $J$ changes sign (**Figure 3A**). The bulk state of the aster could, however, be affected by the sign of $J$, depending on the mode of negative feedback (see Appendix 6 for detailed discussion). When negative feedback promotes depolymerization at high microtubule density, $J < 0$ in the bulk and asters are composed of short microtubules that are created through nucleation and lost through depolymerization. Thus, dynamics are essentially the same as when $J < 0$ both at the front and at the bulk. When negative feedback simply arrests nucleation in the bulk, individual microtubules begin to span the entire aster as in the standard model. The observations of

newly nucleated plus ends during aster growth exclude this latter scenario (*Ishihara et al., 2014a*) .

## Appendix 5

### Microtubule lifetime

Consider a single microtubule nucleated at time $t = 0$. Let $\rho_g(t, l)$ denote the probability that it is of length $l$ at time $t$ and in a growing state. Similarly, $\rho_s(t, l)$ is for the shrinking state. Then,

$$\begin{cases} \dfrac{\partial \rho_g}{\partial t} = -v_g \dfrac{\partial \rho_g}{\partial l} - f_{cat}\rho_g + f_{res}\rho_s, \\[2mm] \dfrac{\partial \rho_s}{\partial t} = +v_s \dfrac{\partial \rho_s}{\partial l} + f_{cat}\rho_g - f_{res}\rho_s, \end{cases} \tag{A36}$$

with initial conditions $\rho_g(t = 0, l) = \delta(l)$ and $\rho_s(t = 0, l) = 0$.

Assuming bounded dynamics $J < 0$, we apply Laplace transforms $t \to s$ and $l \to q$, and solve the problem as before. The result reads

$$\rho_g(s, q) = \frac{1}{v_g} \frac{1}{q - q_-(s)}, \tag{A37}$$

where $q_-$ is the negative root of the quadratic equation (A14).

Now, the average time spent in the growing state $\tau_g$ is given by

$$\begin{aligned} \tau_g &= \int \int \rho_g(t, l) \, dl \, dt \\ &= \rho_g(s = 0, q = 0) \\ &= \frac{-1}{v_g} \left( \frac{v_g f_{res} - v_s f_{cat}}{v_g v_s} \right)^{-1} \\ &= \frac{v_s}{v_s f_{cat} - v_g f_{res}}. \end{aligned} \tag{A38}$$

The last expression above is identical to the inverse of the critical nucleation rate. Thus, $\tau_g r_c = 1$ specifies the equation for the critical nucleation rate, which can be interpreted as the requirement for an average microtubule to nucleate one other microtubule during its lifetime.

Analogously, for the average time spent in the shrinking state $\tau_s$, we find

$$\tau_s = \frac{v_g}{v_s f_{cat} - v_g f_{res}}. \tag{A39}$$

Note that $\frac{\tau_s}{\tau_g} = \frac{v_g}{v_s}$. We can also obtain total lifetime of the microtubule $\tau$ by summing over its lifetimes in the growing and shrinking states:

$$\tau = \tau_g + \tau_s = \frac{v_g + v_s}{v_s f_{cat} - v_g f_{res}}. \tag{A40}$$

This result is identical to that of Bicout (**Bicout, 1997**) who did not consider the lifetimes of growing and shrinking microtubules separately.

## Appendix 6

### Plus end density in the aster interior

In the interior region of a growing aster, the density of microtubule plus ends and microtubule length distribution are stationary and independent of position. As a result, the increase in the number of microtubules due to nucleation must equal microtubule loss. For growing microtubules, the rate of loss is given by $\frac{1}{\tau_g}$ (see **Equation A38**) while the rate of gain is simply the nucleation rate. Thus,

$$r^{bulk} = \frac{1}{\tau_g^{bulk}}. \tag{A41}$$

### Nucleation changes with plus end density

Logistic function is a commonly used mechanism for negative feedback in the context of expanding populations (see (**Korolev, 2013**) for an example). In **Equation (A2)**, it takes the following form

$$r = r_0 \left(1 - \frac{C_g}{K}\right), \tag{A42}$$

where $r_0$ is the nucleation rate at low plus end densities, and $K$ sets the scale of $C_g$ when the negative feedback becomes appreciable, resulting in stationary plus end density. The balance between microtubule production and loss given by **Equation (A41)** results in the following expression for the plus end density in the bulk

$$C_g^b = K\left(\frac{r - r_c}{r}\right), \tag{A43}$$

where we used the fact that $\frac{1}{\tau_g^{bulk}} = r_c$.

Michaelis-Menten type kinetics is an alternative functional form for the negative feedback that could arise, for example, due to the limitation of a nucleating factor,

$$r = \frac{r_0}{1 + \frac{C_g}{K}}. \tag{A44}$$

This results in the following plus end density:

$$C_g^b = K\left(\frac{r - r_c}{r_c}\right). \tag{A45}$$

In either case, $C_g^b$ is proportional to $r - r_c$ close to the transition. Fluctuations in $C_g^b$ due to the stochasticity of microtubule nucleation and collapse can alter this behavior to $(r - r_c)^\beta$, where $\beta$ is the corresponding exponent of a non-equilibrium transition. This transition most likely belongs to the directed percolation universality class (**Hinrichsen, 2000**). Note that the critical nucleation rate used here is the same as in **Equation (5)**. In particular, all the values of all the model parameters are obtained in the limit of small $C_g$, i.e. at the edge of the aster.

## Catastrophe rate changes with plus end density

Instead of changing the nucleation rate, the cell can promote microtubule depolymerization to ensure that the bulk density does not grow indefinitely. Consider the negative feedback such that the nucleation is constant throughout the aster. Specifically, the nucleation term is simply proportional to the local density of growing plus ends as in

$$Q(t,x) = r \cdot C_g(t,x),$$ (A46)

while we implement the feedback regulation in the catastrophe rate,

$$f_{cat} = f_{cat}^0 \left(1 + \frac{C_g}{K}\right).$$ (A47)

Here, $f_{cat}^0$ corresponds to the catastrophe rate at the leading edge of the growing aster where $C_g$ is small and $K$ specifies the plus end densities at which the negative feedback becomes appreciable.

The balance between the constant nucleation rate and the loss rate that increases with the plus end densities leads to the following solution for the steady-state density in the bulk:

$$C_g^b = K \left(\frac{r - r_c}{f_{cat}^0}\right).$$ (A48)

Note that the critical nucleation rate used here is the same as in **Equation (5)**. In particular, all the values of all the model parameters are obtained in the limit of small $C_g$, i.e. at the edge of the aster.

## Depolymerization rate changes with plus end density

As a final example, we consider a situation where the nucleation is constant as in **Equation (A46)**, while the depolymerization rate increases with plus end density,

$$v_s = v_s^0 \left(1 + \frac{C_g}{K}\right).$$ (A49)

Here, $v_s^0$ corresponds to the depolymerization rate at the leading edge of the growing aster where $C_g$ is small. For this feedback mechanism, we find

$$C_g^b = K \left(\frac{r - r_c}{f_{cat} - r}\right).$$ (A50)

In the above four examples we found that close to the onset of aster growth the bulk density is proportional to $r - r_c$, and it is easy to see that this is true regardless of the feedback mechanism. Indeed, at the critical transition, the nucleation barely keeps up with loss at the front; thus, an infinitesimal increase in density and the corresponding negative feedback would alter the balance and $C_g$ must be zero at $r = r_c$. As a result, $C_g^b$ is proportional to $r - r_c$ just above to the transition. In contrast, the expansion velocity near the transition does not vanish and remains at a high value specified by $V_{gap}$. In consequence, the cell can control the density of the microtubules in the aster and, therefore, its mechanical properties by small changes in the nucleation rate without significantly altering the kinetics of aster growth.

## Appendix 7

### Other types of feedback regulation lead to the same explosive transition

Apart from the carrying the capacity on nucleation kinetics considered above, other forms of negative feedback on the system are possible. This may include scenarios such as decreasing polymerization rate or increasing catastrophe rate with a higher local density of microtubules. Feedback regulation at higher microtubule densities is important in the interior of the growing aster while the linearized equations solved above capture the dynamics of the leading edge. Thus, different forms of feedback regulation lead to the same critical transition predicted by *Equations (5) and (A31)*.

Consider one such alternate scenario for (A1) as described in Appendix 6. Catastrophe rate changes with plus end density. Here, the negative feedback is implemented at the level of catastrophe rate instead of nucleation. We numerically solved the partial differential equations under these assumptions. Similar to the previous case, we observed the emergence of propagating fronts in a parameter dependent manner (*Figure 3—figure supplement 1A*). Although the shape of the propagating front is different, the aster velocity is again in excellent agreement with our analytical solution (*Figure 3—figure supplement 1B*).

## Appendix 8

# Estimation of unknown parameters $f_{cat}$ and $r$

By combining analytical solutions and experimental measurements in frog egg extract, we estimate the values of unknown parameters in our model. Given our direct measurements for $v_g$, $v_s$, $f_{cat}$, $V$, and $V_{gap}$, we have two unknowns $f_{cat}$ and $r$. To simultaneously estimate these values, we need two equations.

The first is the aster velocity **Equation (A31)**. For the second equation, we use one of the equations for $V_{gap}$ as shown in **Appendix 4—table 1**, which correspond to different assumptions on how $V_{gap}$ was achieved by the MCAK-Q710 perturbation. The result of the parameter estimations is summarized in **Appendix 8—table 1**. In all scenarios, the values of $f_{res}$ and $r$ are in relative agreement.

**Appendix 8—table 1.** Estimated parameter values for different scenarios on how MCAK-Q710 arrested aster growth. Different expressions for $V_{gap}$ shown in **Appendix 4—table 1** were used. In all cases, the values of $v_g$, $v_s$, $f_{cat}$, $V$, and $V_{gap}$ were the same as in **Table 1**.

| Estimated parameter | Units | $r \rightarrow r_c$ | $f_{res} \rightarrow f_{res,c}$ | $v_s \rightarrow v_{s,c}$ |
|---|---|---|---|---|
| $f_{res}$ | min$^{-1}$ | 2.0±0.3 | 3.0±0.7 | 3.0±0.7 |
| $r$ | min$^{-1}$ | 2.1±0.2 | 1.9±0.2 | 1.8±0.2 |
| $K$ | μm$^{-1}$ | 0.053±0.030 | 0.12±0.09 | 0.15±0.10 |
| $\langle l \rangle$ | μm | 16±2 | 32±34 | 39±44 |

## Appendix 9

### Aster growth by polymer-stimulated nucleation of microtubules

In this Appendix, we consider a scenario where microtubule nucleation is stimulated by the local density of polymer rather than the density of growing plus ends. Although we have not obtained the analytical solution for this scenario, we derive the expression for the critical nucleation rate for aster growth and confirm these results by numerical simulations. Importantly, the transition from stationary to growing asters is predicted to have a finite jump in velocity.

### Critical nucleation rate for polymer-stimulated nucleation

Let $p$ denote the polymer-stimulated nucleation rate with units $[\text{time}^{-1}\,\text{microtubule length}^{-1}]$. A microtubule of length $l$ will nucleate $p\,l\,dt$ microtubules in time $dt$.

To derive the critical nucleation rate $p_c$ for aster growth, we require that a single microtubule during its entire lifetime must nucleate at least one microtubule:

$$p_c \int_0^\infty dt \int_0^\infty dl\, l\, \rho(t,l) = 1, \tag{A51}$$

where $\rho(t,l)$ denotes the local density of all plus ends. As plus ends are either in the growing or shrinking states, $\rho(t,l) = \rho_g + \rho_s$.

By using Laplace transforms in the left hand side of **Equation (A51)**, we obtain

$$\int_0^\infty dt \int_0^\infty dl\, l\, \rho(t,l) = \int_0^\infty \rho(s=0,l)\, l\, dl = -\frac{d\rho(s=0,q)}{dq}\Big|_{q=0}. \tag{A52}$$

Thus, we may solve for the desired critical nucleation rate as $p_c = \left( -\frac{d\rho(s=0,q)}{dq}\big|_{q=0} \right)^{-1}$.

We have previously obtained the expression for $\rho_g$ in **Equation (A37)**. To obtain the equivalent expression for $\rho_s$, we return to the dynamic equation **Equation (A36)** and apply the Laplace transforms $t \to s$ and $l \to q$,

$$s\rho_g - 1 = -v_g q \rho_g - f_{cat}\rho_g + f_{res}\rho_s, \tag{A53}$$

which yields,

$$\rho_s(s,q) = \frac{\rho_g(s + v_g q + f_{cat}) - 1}{f_{res}}. \tag{A54}$$

Combining **Equation (A37) and (A54)**, we obtain

$$\rho(s,q(s)) = \rho_g + \rho_s = -\frac{1}{f_{res}} + \frac{s + v_g q(s) + f_{cat}}{v_g f_{res}} \cdot \frac{1}{q(s) - q_-} + \frac{1}{v_g} \cdot \frac{1}{q(s) - q_-}. \tag{A55}$$

Differentiating and setting $q = 0$, we find

$$-\frac{d\rho(s=0,q)}{dq}\Big|_{q=0} = \frac{1}{v_g f_{res}} \cdot \frac{v_g q_- + f_{cat}}{q_-^2} + \frac{1}{v_g} \cdot \frac{1}{q_-^2}$$
$$= \frac{1}{v_g q_-^2}\left(1 + \frac{f_{cat}}{f_{res}}\right) + \frac{1}{f_{res} q_-}. \tag{A56}$$

From *Equation (A14)*, we find

$$q_-(s=0) = -\frac{v_s f_{cat} - v_g f_{res}}{v_g v_s}. \tag{A57}$$

Using *Equation (A56) and (A57)*, we solve for the critical nucleation rate

$$p_c = \left(-\frac{d\rho(s=0,q)}{dq}\Big|_{q=0}\right)^{-1}$$
$$= \frac{f_{res} v_g q_-^2}{f_{res}\left(1 + \frac{f_{cat}}{f_{res}}\right) + v_g q_-}$$
$$= f_{res} v_g \cdot \frac{(v_s f_{cat} - v_g f_{res})^2}{v_g^2 v_s^2} \cdot \frac{1}{f_{res} + f_{cat} - f_{cat} + \frac{v_g}{v_s} f_{res}}$$
$$= \frac{(v_s f_{cat} - v_g f_{res})^2}{v_g v_s (v_g + v_s)}. \tag{A58}$$

In the scenario of polymer-stimulated nucleation, the minimal nucleation rate required for aster growth is

$$p_c = \frac{(v_s f_{cat} - v_g f_{res})^2}{v_g v_s (v_g + v_s)}. \tag{A59}$$

When both types of nucleation are present, we expect $\frac{r}{r_c} + \frac{p}{p_c} = 1$ to define the transition, where $r_c$ and $p_c$ are defined in the absence of the other type of nucleation as in *Equation (5) and (A59).*

## Numerical simulations predict a gap velocity for aster growth by polymer-stimulated nucleation

We modified our numerical simulation to ask if polymer-stimulated nucleation predicts the aster growth. Similar to the scenario of growing-plus-end-stimulated nucleation (*Figure 2A*), low nucleation rate predicts a stationary aster (*Figure 3—figure supplement 2A*, left), while high nucleation rate predicts an aster that continuously increases in radius (*Figure 3—figure supplement 2A*, right) even when individual microtubules are unstable ($J<0$). To systematically explore the polymer-stimulated nucleation scenario, we varied the model parameters and measured the aster growth velocity $V$. We find that the transition from a stationary to a growing aster is accompanied by a finite jump in $V$ (*Figure 3—figure supplement 2B and C*). Our predictions for the critical polymer nucleation rates $p_c$ is in excellent quantitative agreement.

## Comparison of autocatalytic nucleation mechanisms and predictions for aster growth

Here, we compare and summarize the theoretical predictions of growing-plus-end-stimulated nucleation vs. polymer-stimulated nucleation. Both scenarios predict

1. stationary and continuously growing asters in a parameter dependent manner

2. the feasibility of aster growth with $J<0$, and that such asters are composed of short microtubules

3. explosive transition to growth, or 'gap velocity', which allows independent control of aster density and growth velocity.

The two scenarios predict qualitatively different transitions when the nucleation rate is increased and the aster velocity reaches $V = v_g$. With growing-plus-end stimulated nucleation, $V$ approached $v_g$ in a smooth manner (**Figure 3**). In contrast, the polymer-stimulated nucleation predicted a finite jump of $V$ to $v_g$ (**Figure 3—figure supplement 2A**, right). In the future, it may be possible to exploit this difference to distinguish the two scenarios of nucleation experimentally.

## Appendix 10

### Gap velocity constrains possible models of aster growth

Here, we describe three examples of potential aster growth models that are readily rejected by the existence of a gap velocity. In all cases, let us assume that microtubules are unstable (bounded dynamics $J<0$) with finite lifetime. Note that all three models do not account for the internal dynamics of agents, namely, the length information of individual microtubules.

#### A simple expanding shell model with autocatalytic nucleation

Consider an aster growth model that does not keep track of microtubule positions, but simply translates the number of microtubules into aster size:

$$\frac{dN_+}{dt} \sim \left(r - \frac{1}{\tau}\right)N_+^\alpha$$
$$R^d \sim N_+ \tag{A60}$$

$N_+$ is the number of plus ends, $R$ is the aster radius, $d$ is the number of spatial dimensions, $r$ is the nucleation rate, and $\tau$ is the microtubule lifetime. $\alpha = 1$ corresponds to no negative feedback, while $\alpha = \frac{d-1}{d}$ corresponds to nucleation only at aster periphery. This is the simplest, virtually non-spatial model. For $\alpha = 1$, the growth is exponential in time rather than linear. For $\alpha = \frac{d-1}{d}$, the growth is linear with the velocity $V \sim \left(r - \frac{1}{\tau}\right)$. Thus, the model exhibits critical nucleation (i.e. both stationary and growing asters), but no gap velocity.

#### A reaction-diffusion model

Previously, we hypothesized a Fisher-Kolmogorov type, reaction-diffusion model of aster growth focusing on plus end dynamics (*Ishihara et al., 2014b*) and autocatalytic nucleation. Denoting the plus end density as $C_+$, carrying capacity as $K$, and the effective growth rate as $r - \frac{1}{\tau}$, the model is as follows:

$$\frac{\partial C_+}{\partial t} = D\frac{\partial^2 C_+}{\partial x^2} + \left(r - \frac{1}{\tau}\right)C_+\left(1 - \frac{C_+}{K}\right) \tag{A61}$$

This predicts an aster velocity of $V \sim \sqrt{D\left(r - \frac{1}{\tau}\right)}$ for $r > \frac{1}{\tau}$ with no gap velocity.

#### A reaction-diffusion model with cooperative nucleation

A more general reaction-diffusion model is obtained by replacing the logistic growth term in *Equation (A61)* by an arbitrary nonlinear function of the plus end density $F(C_+)$:

$$\frac{\partial C_+}{\partial t} = D\frac{\partial^2 C_+}{\partial x^2} + F(C_+) \tag{A62}$$

$F(C_+)$ can specify whether there is a minimal concentration of microtubules necessary for nucleation or account for other effects such as cooperativity. Despite the possibilities of quite complicated nucleation dynamics, all reaction-diffusion models specified by *Equation (A62)* exhibit no gap velocity (*Murray, 2002*). Gap velocity has also not been observed in a variety of further extensions of *Equation (A62)* that account for advection terms and density-dependent diffusion (*Murray, 2002*).

