## [Decision Letter]

Thank you for submitting your article "Physical basis of large microtubule aster growth" for consideration by *eLife*. Your article has been reviewed by two peer reviewers, and the evaluation has been overseen by a Reviewing Editor and Arup Chakraborty as the Senior Editor. The reviewers have opted to remain anonymous.

The reviewers have discussed the reviews with one another and the Reviewing Editor has drafted this decision to help you prepare a revised submission.

Summary:

Your paper describing a quantitative biophysical model of aster growth with autocatalytic nucleation has been seen by two reviewers. We feel this new model for growth of large microtubule asters is a major advance for the microtubule field. It also will have an important impact on conceptual thinking about polymer dynamics in the cytoskeletal field. Both reviewers, who although making different points, reach the same overall conclusion, which is that many of the assumptions are not sufficiently explored or described so as to be able to constrain your model. While the prediction from the model is straightforward (*V_gap_*), the experimental validation is often not sufficient to test the predictions.

Essential revisions:

The applicability of the Equation for *J* in Figure 1 for nucleated microtubule dynamics needs to be better clarified early in the Results. In a number of places in the manuscript, it is stated that on average the microtubules are "shrinking". Yet, collectively the aster is growing enormously by the mechanism modeled here. So how can microtubules be on average shrinking while the aster is growing? By the Equation of *J* they are on average "shrinking", but since all microtubules within the expanded aster are nucleated, why is the Equation for *J* in Figure 1 applicable to this analysis? The equation in Figure 1 does not include the influence of the nucleation site in stopping depolymerization; how is this geometrical constraint implemented in the analysis?

One reviewer noted, "Here is an example of my confusion. Suppose there is nucleated microtubule growth, no spontaneous rescue and rescue occurs only by depolymerization to the initial nucleation site. Say *V_g_* = 20 um/min, *t_g_* = 1 min, and *V_s_* = 200 um/min. So the plot of MT dynamic instability is almost saw-toothed because *V_s_* >> *V_g_*. So, on a time average, the microtubules are mainly growing, because shortening is so fast. However, the value of *J* = 0, because length gained in growth is always lost in shortening. If the constraints from nucleation are neglected, with no spontaneous rescue, the value of *J* from the equation in Figure 1 is <<0, which is of course a lot of shrinking".

Also with respect to nucleation, it is represented by the term *Q*, which depends on the nucleation rate *r*, the carrying capacity *K* and the local density of growing microtubule plus ends *C_g_*. There are several problems with this definition:

*r* is not measured / cannot be measured, but interfered with the aster growth velocity equation.

*K* is estimated by comparing Cgbulk from experiments to predicted values.

*C_g_* and Cgbulk are both derived from EB1 tracks, which are growing microtubule ends. This term should be changed to the local density of all microtubules rather than growing microtubule plus ends. Or is there any evidence that microtubule-dependent microtubule nucleation exclusively happens on growing microtubules?

The reviewers feel that there is not sufficient discussion of parameters that have not been measured. For instance, "We explored perturbation of various dynamics regulators, seeking one whose effect was restricted to influencing a single parameter in our model. This was challenging since proteins that regulate polymerization dynamics tend to also regulate nucleation…".

In the paper, you state that the only dynamic parameter that has been actually quantified is *f_cat_*. This seems a bit misleading, as *f_cat_* was not measured directly. We feel that, all dynamic parameters should be measured to be sure that treatment with MCAK-Q710 indeed only changes *f_cat_*, or if not, an explanation as to why not.

*f_cat_* is the predominant dynamic parameter of microtubules that has the power to dramatically change microtubule behavior. Therefore, the authors need to reliably quantify *f_cat_* by an appropriate method (e.g. speckle microscopy) and in addition measure all other dynamic parameters.

In addition, the reference given is misleading: In "Tirnauer, J. S., Salmon, E. D., and Mitchison, T. J. (2004). Microtubule plus-end dynamics in *Xenopus* egg extract spindles. Molecular biology of the cell, 15(4):1776{1784}." Microtubule growth and depolymerization velocity is measured, but not *f_cat_*.

---

## [Author Response]

*Essential revisions:*

*The applicability of the Equation for J in Figure 1 for nucleated microtubule dynamics needs to be better clarified early in the Results. In a number of places in the manuscript, it is stated that on average the microtubules are "shrinking". Yet, collectively the aster is growing enormously by the mechanism modeled here. So how can microtubules be on average shrinking while the aster is growing? By the Equation of J they are on average "shrinking", but since all microtubules within the expanded aster are nucleated, why is the Equation for J in Figure 1 applicable to this analysis? The equation in Figure 1 does not include the influence of the nucleation site in stopping depolymerization; how is this geometrical constraint implemented in the analysis?*

*One reviewer noted, "Here is an example of my confusion. Suppose there is nucleated microtubule growth, no spontaneous rescue and rescue occurs only by depolymerization to the initial nucleation site. Say V_g_ = 20 um/min, t_g_ = 1 min, and V_s_ = 200 um/min. So the plot of MT dynamic instability is almost saw-toothed because V_s_ >> V_g_. So, on a time average, the microtubules are mainly growing, because shortening is so fast. However, the value of J = 0, because length gained in growth is always lost in shortening. If the constraints from nucleation are neglected, with no spontaneous rescue, the value of J from the equation in Figure 1 is <<0, which is of course a lot of shrinking".*

The reviewers are bringing up several important points in these comments:

1) How can aster grow when *J*<0 and microtubules tend to shrink on average?

2) What is the meaning of *J* in a model with nucleation compared to the model with the model without nucleation? How does this relate to aster growth velocity *V*?

3) What happens when microtubules depolymerize completely and does this impose a geometric constraint?

We had, to some extent, addressed these questions in the original submission, but it was clear from this set of reviewer comments that we had not done a good enough job in explaining our terms, assumptions and reasoning. We address these comments below first highlighting the general picture and then answering specific comments. We have added significantly more explanatory text in the revised submission. We thank the reviewer for asking these questions, since our revisions in response have clearly improved the manuscript.

1) How do asters grow when *J*<0 and microtubules tend to depolymerize on average?

We had not defined the difference between microtubule growth and aster growth in the previous version. We now define, early in the results, aster growth rate as *V*, which is a collective property of all the microtubules, and average growth rate of individual microtubules, *J*, which we define the same way as Dogterom and Leibler and subsequent authors.

Whether or not asters can grow (*V*>0) when individual microtubules are unstable (*J*<0) was the single most important question we sought to address in our study. We suspected they could, and our theory work confirms this surprising idea. We evidently failed to adequately explain our conclusions, and have made significant text changes to do a better job.

First, we added the following sentence at the beginning of the Results section to clearly state the motivations for the theory:

“Asters are large structures comprised of thousands of microtubules. How do the microscopic dynamics of individual microtubules determine the collective properties of asters such as their morphology and growth rate? […] Polymerization dynamics of plus ends is an individual property of microtubules.”

Next, we provide an intuitive picture of aster growth by use of analogy. In some ways, aster growth resembles the growth of the population size of a species. Each individual is mortal and destined to die, yet the population size grows because reproduction outweighs death. Similarly, the aster will grow, if, on average, a single microtubule nucleates more than one microtubule before it shrinks to the nucleation site, an event that removes both the microtubule and the nucleation site from the system. Due to the geometry of the aster, and birth of new microtubule near the periphery, this increase will cause the aster radius to grow even if individual microtubules are unstable.

Below is the text that we added to provide an intuitive picture of the collective growth regime:

“In this regime, asters grow (*V*>0) although individual microtubules are bounded (*J*<0) and are, therefore, destined to shrink and disappear. […] As we show below, this self-amplifying propagation of microtubules is possible only for sufficiently high nucleation rates necessary to overcome microtubule loss and sustain collective growth.”

We also modified the figure legend for the phase diagram of aster growth in Figure 5 to better integrate how *J* and *V* relate to each other when nucleation is considered.

2) What is the definition and meaning of *J*? How does this relate to aster growth velocity *V*?

First, let us point out that the definition and meaning of *J* are the same with and without nucleation because *J* describes the dynamics of individual microtubules. In particular, *J* is not the velocity of aster expansion in general. *V* and *J* are quite different, and the point of our model is to examine the relationship between them.

We define *J* following Dogterom and Leibler (1993), PRL and Verde et al. (1992), JCB: J= vg fres− vsfcatfres  +fcat both for r=0 and r>0. We now introduce this equation as an equation in the main text.

One useful interpretation of this mathematical construct is the following. Consider an ensemble of very, very long microtubules, *f_res_/(f_res_* +*f_cat_*) of which are in the growing state and *f_cat_/(f_res_*+*f_cat_*) of which are in the shrinking state. Then, the mean length of microtubules in this ensemble changes with a rate equal *J*. Two things are important for this interpretation. First, we assumed very long microtubules so that we can neglect the effects of disappearance and re-nucleation upon shrinking. Second, we assumed that the fraction of growing and shrinking microtubules are given by their steady state fractions (*f_res_/(f_res_*+*f_cat_*) and *f_cat_/(f_res_*+*f_cat_*)). These assumptions do not hold during aster growth and care needs to taken when interpreting *J* intuitively. A very similar interpretation of *J* appears when one attempts to describe the dynamics of microtubule length as a random walk. In this case, *J* is the mean velocity, or bias, of the random walk.

Without an explaining the assumptions as mentioned above, referring to *J* as the mean polymerization rate may cause confusion. Indeed, each microtubule eventually disappears, which implies that the mean increase in length is exactly zero. To understand the apparent paradox, note that each new microtubule is nucleated in growing state; therefore, its polymerization rate is positive at early times even when *J*<0. The value of *J* then quantifies the bias towards polymerization vs. depolymerization rather the change is the mean aster length. In particular, for *J*<0, microtubules collapse faster and remain shorter the greater the magnitude of this negative bias. Thus, when we said “microtubules shrink on average,” we meant that there is a bias towards depolymerization. We removed this language from the manuscript and replaced it with “unstable microtubules”, “bounded microtubules”, or simply “*J*<0”.

Further, we substantially modified the paragraph describing “*J*>0” and “*J*<0” to better explain our definition of *J* and also describe how *J* and *V* are related in the standard model:

“Plus end dynamics can be conveniently summarized by the time-weighted average of the polymerization and depolymerization rates (Dogterom and Leibler, 1993; Verde et al., 1992):

J= vg fres− vsfcatfres  +fcat

[…] With unbounded dynamics *J*>0, the standard model predicts an aster that has a constant number of microtubules and increases its radius at a rate equal to the elongation rate of microtubules (i.e. *V*=*J*).”

We also concluded that the presentation of our model, a coupled partial differentiation equation, was not accessible to the general audience. We added the following sentence to supplement our equations to emphasize that our mathematical model incorporates the established concept of microtubule dynamics described above:

“To understand this system of equations, consider the limit of no nucleation (*r*=0). Then, the equations at different *x* become independent and we recover the standard model that reduces aster growth to the growth of individual microtubules (Dogterom and Leibler, 1993; Verde et al., 1992). With nucleation, aster growth is a collective phenomenon because microtubules…”

Finally, to highlight that the aster growth velocity *V* is a collective property resulting from the interplay of polymerization dynamics and nucleation, we now feature the analytical solution for *V* in the Results section (formerly in Methods).

3) How do nucleation sites affect microtubule dynamics?

In our analysis there are two types of nucleation events: one from centrosomes and the other in the cytoplasm stimulated by microtubules. In simulations, we treat them differently. For centrosomal microtubules, we assume immediate re-nucleation because centrosomes serve as potent nucleating centers. For the microtubules nucleated elsewhere in the cytoplasm, we assume that the microtubule collapses and disappears. Thus, the only geometric constraint is at the centrosome. In the mathematical analysis, we ignore centrosomal microtubules since they do not extend far into the aster, and the aster growth is driven exclusively by non-centrosomal nucleation. We have now added a paragraph to the main text explicitly discussing this assumption:

“Finally, we need to specify what happens when microtubules shrink to zero length. In our model, microtubules originating from centrosomes rapidly switch from shrinking to growth (i.e. re-nucleate), while non-centrosomal microtubules disappears completely (i.e. no re-nucleation occurs). […] As a result, minus ends need to be stabilized after nucleation possibly by some additional factors (Akhmanova and Hoogenraad, 2015) and mechanical integrity of the aster should rely on microtubule bundling (Ishihara et al., 2014a).”

Finally, we want to further clarify our response to the following comment by the reviewer:

*"Here is an example of my confusion. Suppose there is nucleated microtubule growth, no spontaneous rescue and rescue occurs only by depolymerization to the initial nucleation site. Say V_g_ = 20 um/min, t_g_ = 1 min, and V_s_ = 200 um/min. So the plot of MT dynamic instability is almost saw-toothed because V_s_ >> V_g_. So, on a time average, the microtubules are mainly growing, because shortening is so fast. However, the value of J = 0, because length gained in growth is always lost in shortening. If the constraints from nucleation are neglected, with no spontaneous rescue, the value of J from the equation in Figure 1 is <<0, which is of course a lot of shrinking".*

All of the reviewer’s predictions are correct. There are indeed several nonequivalent ways to characterize the tendency to shrink: time spent shrinking, total change in distance, bias towards shrinking *J*, etc. We also point out that the example given above equally applies to the standard theory with only centrosomal nucleation, where the concept of *J* was first introduced. Our motivation for using *J* rather than any other quantity is that *J*=0 marks an important transition. For *J*>0, microtubules expand forever upon nucleation, while, for *J*<0, microtubules eventually shrink and need to be re-nucleated. The comment of the reviewer also highlights the dynamic complexity of aster growth and the need for rigorous mathematical analysis to supplement one’s intuition. The exact solution presented in our manuscript addresses precisely this problem.

*Also with respect to nucleation, it is represented by the term Q, which depends on the nucleation rate r, the carrying capacity K and the local density of growing microtubule plus ends C_g_. There are several problems with this definition:*

*r is not measured / cannot be measured, but interfered with the aster growth velocity equation.*

We think that introducing nucleation rate in the term describing nucleation is unavoidable and any definition must have this problem. By including terms in the model that we know exist but cannot yet quantify or define precisely, we made novel predictions that one would not expect if not for the model. Importantly, our main predictions require hypothesizing only a single parameter *r* for the autocatalytic nucleation, and is fairly robust to the exact form of the autocatalytic nucleation term (SI) and exclude alternative.

Theory is valuable precisely because there are no direct methods to measure nucleation rate or even observe nucleation.

As described in our response below, we found that neither *V_g_* nor *f_cat_* changed with increasing MCAK-Q710 concentrations, which transformed growing asters to stationary ones (Figure 4). This is consistent with the perturbation altering *V_s_, f_res_*, and/or *r*. Given these possibilities, we now provide a dedicated explanation in the Discussion for how our equations for aster growth velocity *V* and gap velocity were combined with experimental measurements to infer the rescue rate and nucleation rate:

“Our biophysical model offers a biologically appealing explanation to aster growth and allows us to estimate parameters that are not directly accessible: the rescue and autocatalytic nucleation rates. […] Thus, our model recapitulates aster growth with reasonable parameter values and offers a new understanding for how asters grow to span large cytoplasms even when individual microtubules are unstable.”

We have greatly re-organized and re-evaluated the parameter estimation for the updated Table 1. Further, we provide the new Appendix 8 and Appendix 8—table 2, which compares the alternative scenarios for how MCAK-Q710 could have arrested aster growth.

We hope we have conveyed the value of combining theory and measurement. We also thank the reviewers for suggesting additional measurements in MCAK-Q710 condition.

*K is estimated by comparing*
Cgbulk
*from experiments to predicted values.*

This seems to be the most direct measurement of this model parameter other than directly observing how the nucleation rate changes with microtubule density. The asters have much lower microtubule density compared to structures such as the mitotic spindle and the low signal to noise of the tubulin fluorescence intensity is unreliable to estimate the microtubule polymer density. In contrast, EB1 counting provides absolute numbers of growing plus ends with considerable sensitivity. We tried quite hard to use electron microscopy to count MTs, but were unsuccessful at fixing the asters while maintaining their structure.

*C_g_ and*
Cgbulk
*are both derived from EB1 tracks, which are growing microtubule ends. This term should be changed to the local density of all microtubules rather than growing microtubule plus ends. Or is there any evidence that microtubule-dependent microtubule nucleation exclusively happens on growing microtubules?*

The reviewer raises an important issue about autocatalytic microtubule nucleation: plus end stimulated vs. polymer stimulated. Based on the literature, we think that both mechanisms are equally appealing and speculative at this point. There are many proteins that bind laterally to microtubules which may target and activate a nucleation complex (e.g. the Augmin complex, which we have depleted in this system and found normal aster growth). The polymerizing plus end (with its GTP cap) is also known to specifically recruit special proteins. Both scenarios currently lack definite molecular or structural evidence. In our original manuscript, we focused on the scenario of plus-end-stimulated nucleation from numerical simulations and analytical solutions.

Encouraged by the reviewer’s comment, we explored the alternative scenario of polymer-stimulated nucleation. Numerical simulations and our newly obtained analytical solution for the critical nucleation rate together confirm that this scenario recapitulates the qualitative features of the aster growth transition similar to the plus-end-stimulated scenario, including the finite jump in aster growth velocity before arrest. We present these results in a dedicated section in the Appendix (Appendix 9) and provide Figure 3—figure supplement 2.

To refer to these results, we use Discussion section to consider the two different scenarios for autocatalytic nucleation. We add the following:

“To date, few studies have rigorously compared the mechanistic consequences of plus-end-stimulated vs. polymer-stimulated nucleation. Above, we presented the theoretical predictions for aster growth by plus-end stimulated nucleation. […] This difference has important implications for the structural mechanism of microtubule nucleation and for the prediction of cell-scale phenomena. For the issue of large aster growth, we propose specific experiments that might be able distinguish these scenarios (Appendix).”

By responding to the reviewer comment and analyzing the alternative scenario of polymer-stimulated nucleation, we feel that we have made an advance in our understanding in the intricacies of autocatalytic nucleation, which remains an unsolved question for the field.

*The reviewers feel that there is not sufficient discussion of parameters that have not been measured. For instance, "We explored perturbation of various dynamics regulators, seeking one whose effect was restricted to influencing a single parameter in our model. This was challenging since proteins that regulate polymerization dynamics tend to also regulate nucleation…".*

We apologize for not providing sufficient discussion of the model parameters. In particular, we failed to emphasize that we had measured (1) *V_g_* and *V_s_* via direct imaging of microtubules in our previous paper (Ishihara et al., PNAS 2014) and (2) *V_g_* and *f_cat_* via EB1 comet tracking in the current report. This left us with *f_res_* and *r* as two unknown parameters which we estimated by combining our theoretical results and experimental measurements aster growth velocity *V* and *V_gap_*. Please also refer to our third response above and the revised Table 1 which summarizes the parameter values.

*In the paper, you state that the only dynamic parameter that has been actually quantified is f_cat_. This seems a bit misleading, as f_cat_ was not measured directly. We feel that, all dynamic parameters should be measured to be sure that treatment with MCAK-Q710 indeed only changes f_cat_, or if not, an explanation as to why not.*

*f_cat_ is the predominant dynamic parameter of microtubules that has the power to dramatically change microtubule behavior. Therefore, the authors need to reliably quantify f_cat_ by an appropriate method (e.g. speckle microscopy) and in addition measure all other dynamic parameters.*

The main point we wanted to test with the MCAK-Q710 experiment was qualitative. Our theory predicts that any combination of changes in parameters can transform the growing aster to a stationary one and that this transition is marked by a finite jump in the aster growth velocity (i.e. gap velocity). We modified the main text to better convey the intent of this experiment with less focus on the need to change a single parameter. Our MCAK-Q710 experiments confirmed the expected qualitative behavior. We added the following sentence to relate the theory and experimental observation:

“In our model, this behavior is consistent with any change of parameter(s) that reduces the aster growth velocity (Eq. (4)) and arrests growth.”

In the literature, wild type MCAK is characterized as positive regulator of catastrophe events. Further, as the reviewer pointed out, *f_cat_* is a critical parameter for microtubule dynamics. However, it was unclear which parameter(s) were actually affected by MCAK-Q710 during the transition.

We performed additional EB1 tracking experiments in the growing aster to measure *V_g_* and *f_cat_* as a function of MCAK-Q710 concentration. The results are presented as a new figure (Figure 4—figure supplement 3).

In summary, we found that neither *V_g_* or *f_cat_* were significantly altered with increasing MCAK-Q710. This was a rather surprising result, considering the wild type protein’s documented role in promoting catastrophe events. At this point, we can only speculate that MCAK-Q710 altered *f_res_*, *V_s_*, and/or the nucleation rate r to arrest aster growth in our experiments (Figure 4). Our speculation is consistent with the previous report that overexpression of MCAK-Q710 in cell lines found a significant reduction of tubulin polymer level (Ayana & Wordeman, Biochem. J. 2004). To incorporate these new findings in the manuscript, we add the following paragraph to end our Results section:

“Finally, we asked which parameters in our model were altered in the MCAK-Q710 perturbation. […] Our data is consistent with the following three scenarios for how MCAK-Q710 antagonizes microtubule assembly: (i) increased depolymerization rate, (ii) decreased rescue rate, and/or (iii) decreased nucleation rate.”

Finally, we respond to the reviewer’s suggestion of speckle microscopy. Although single molecule lifetime imaging (via speckle microscopy) offers a timescale of tubulin turnover, it gives only one value and requires additional measurements or assumptions to constrain any of the four parameters in the two state model.

Even if we could measure all four parameters, it does not significantly add to our main observation of gap velocity. Thus, we chose to focus on the EB1-based analysis to directly address if and how *V_g_* and *f_cat_* changes in the MCAK-Q710 perturbation. We hope that our experiments and reasoning addressed the reviewers’ concerns with the MCAK-Q710 experiment.

*In addition, the reference given is misleading: In "Tirnauer, J. S., Salmon, E. D., and Mitchison, T. J. (2004). Microtubule plus-end dynamics in Xenopus egg extract spindles. Molecular biology of the cell, 15(4):1776{1784}." Microtubule growth and depolymerization velocity is measured, but not f_cat_.*

Thank you. We’ve changed the reference to one that used EB1 trajectories to estimate catastrophe rate: Matov et al., 2010, Nature Methods, Analysis of microtubule dynamic instability using a plus-end growth marker.